# Identification of TNFAIP2 as a unique cellular regulator of CSF-1 receptor activation

Randa A Abdelnaser[1], Masateru Hiyoshi[2], Naofumi Takahashi[1], Youssef M Eltalkhawy[1], Hidenobu Mizuno[3], Shunsuke Kimura[4], Koji Hase[4], Hiroshi Ohno[5], Kazuaki Monde[6], Akira Ono[7], Shinya Suzu[1]

The receptor of CSF-1 (CSF1R) encoding tyrosine kinase is essential for tissue macrophage development, and the therapeutic target for many tumors. However, it is not completely understood how CSF1R activation is regulated. Here, we identify the cellular protein TNF-$\alpha$–induced protein 2 (TNFAIP2) as a unique regulator of CSF1R. CSF1R forms large aggregates in macrophages via unknown mechanisms. The inhibition or knockdown of TNFAIP2 reduced CSF1R aggregate formation and functional response of macrophages to CSF-1, which was consistent with reduced CSF1R activation after CSF-1 stimulation. When expressed in 293 cells, TNFAIP2 augmented CSF1R aggregate formation and CSF-1–induced CSF1R activation. CSF1R and TNFAIP2 bind the cellular phosphatidylinositol 4,5-bisphosphate (PIP2). The removal of the PIP2-binding motif of CSF1R or TNFAIP2, or the depletion of cellular PIP2 reduced CSF1R aggregate formation. Moreover, TNFAIP2 altered the cellular distribution of PIP2. Because CSF-1–induced dimerization of CSF1R is critical for its activation, our findings suggest that TNFAIP2 augments CSF1R aggregate formation via PIP2, which brings CSF1R monomers close to each other and enables the efficient dimerization and activation of CSF1R in response to CSF-1.

## Introduction

Macrophages are innate immune cells that orchestrate homeostasis, inflammation, or regeneration, and present across various tissues throughout the body (1, 2). The pool of macrophages in each tissue is maintained by the differentiation of bone marrow–derived monocytes and/or the self-renewal of macrophages originated from the precursors in the extraembryonic yolk sac or monocytes in the fetal liver (3). The development, proliferation, and survival of tissue macrophages can be regulated by cytokines, such as CSF-1, IL-34, and CSF-2 (4). CSF-1 (also known as M-CSF) and IL-34 share the receptor CSF1R (also known as Fms) (5, 6), and mice lacking CSF1R are deficient in most tissue macrophages (7). Interestingly, mice lacking CSF-1 are deficient in most tissue macrophages, but not in Langerhans cells and microglia, whereas mice lacking IL-34 exhibit a selective reduction in Langerhans cells and microglia, which is because CSF-1 is ubiquitously expressed whereas IL-34 is mainly expressed by keratinocytes and neurons (8, 9). Meanwhile, mice lacking CSF-2 (also known as GM-CSF) show functional defects in alveolar macrophages but no major deficiency in most tissue macrophages (10, 11). Thus, the CSF-1/IL-34/CSF1R axis is critical for the development of most tissue macrophages.

Macrophages are abundant immune cells in tumor microenvironment, and the CSF-1/IL-34/CSF1R axis often induces pro-tumorigenic macrophages (12). Thus, CSF1R is one of the attractive therapeutic targets for a variety of tumors. For instance, the small molecule CSF1R inhibitor BLZ945 blocked the progression of glioma in preclinical models (13), and the anti-CSF1R antibody RG7155 reduced tumor burden in patients with diffuse-type giant cell tumors (14). Many clinical studies targeting CSF1R are ongoing (15).

CSF1R is a receptor tyrosine kinase and belongs to class III receptor tyrosine kinase that includes Flt3, Kit, PDGFR$\alpha$, and PDGFR$\beta$ (16, 17). In the absence of CSF-1 or IL-34, CSF1R is present as a monomer and the CSF1R monomer is inactive because of cis-autoinhibition (16, 17). The binding of CSF-1 or IL-34, both of which are homodimers, induces the dimerization of CSF1R and releases the cis-autoinhibition (16, 17, 18). The change leads to the activation and autophosphorylation of CSF1R, which results in the tyrosine phosphorylation of downstream proteins and activation of various signaling pathways, including MAP kinases (6, 17, 19, 20). Thus, the dimerization is critical for the subsequent activation and autophosphorylation of CSF1R. However, it is not completely understood how CSF1R dimerization is regulated. Interestingly, the biochemical analysis raised the possibility that CSF1R monomers are clustered and form aggregates in the mouse macrophage cell

[1]Joint Research Center for Human Retrovirus Infection, Kumamoto University, Kumamoto, Japan [2]Research Center for Biological Products in the Next Generation, National Institute of Infectious Diseases, Tokyo, Japan [3]International Research Center for Medical Sciences, Kumamoto University, Kumamoto, Japan [4]Division of Biochemistry, Faculty of Pharmacy, Keio University, Tokyo, Japan [5]Laboratory for Intestinal Ecosystem, RIKEN Center for Integrative Medical Sciences, Yokohama, Japan [6]Department of Microbiology, Faculty of Life Sciences, Kumamoto University, Kumamoto, Japan [7]Department of Microbiology and Immunology, University of Michigan Medical School, Ann Arbor, MI, USA

Correspondence: ssuzu06@kumamoto-u.ac.jp

line BAC1.2F5 (21). Consistent with this, CSF1R was detected as aggregate-like signals in BAC1.2F5 cells in the immunofluorescence analysis (22, 23). Meanwhile, CSF1R expressed in the breast cancer cell line SKBR3 did not form such aggregates, but CSF-1 could stimulate their proliferation (23). Thus, the formation of CSF1R aggregates may not be an absolute requirement for its ligands to activate CSF1R, but the pre-formed CSF1R aggregates, in which the monomers are close to each other in macrophages, may be beneficial for CSF-1 or IL-34 to dimerize and activate CSF1R (21). However, to what extent CSF1R aggregate formation contributes to dimerization/activation of CSF1R remains unexplored. Furthermore, little is known about the molecular mechanism by which CSF1R forms the aggregates.

In this study, we show that a cellular protein TNF-$\alpha$–induced protein 2 (TNFAIP2; also known as M-Sec or B94) is involved in the CSF1R aggregate formation and contributes to an efficient activation of CSF1R in macrophages. *TNFAIP2*, which was initially identified as a TNF-$\alpha$–inducible gene in endothelial cells (24), is highly expressed in hematopoietic tissues (25) and enriched in myeloid cells, such as neutrophils, dendritic cells, monocytes, and macrophages (26). TNFAIP2 is the 74-kD cytosolic protein with no known enzymatic activity and shares a homology with Sec6 (26), a component of the exocyst complex. The well-known function of TNFAIP2 is the formation of tunneling nanotubes (26, 27), the F-actin–containing long plasma membrane protrusions. TNFAIP2 is also known to enhance cell motility (28, 29). Interestingly, podocytes, the glomerular visceral epithelial cells, also constitutively express TNFAIP2, and the TNFAIP2 knockout mice develop focal segmental glomerulosclerosis (30). In addition, we have demonstrated that TNFAIP2 facilitates cell-to-cell transmission of human retroviruses, such as HIV-1 and HTLV-1 (31, 32, 33). For instance, the TNFAIP2 inhibitor NPD3064, a small chemical that inhibits TNFAIP2-mediated tunneling nanotube formation (31), reduced the production of HIV-1 in macrophages (31). These functions of TNFAIP2 might be at least in part because of the formation of tunneling nanotubes (30, 31). However, physiological functions of TNFAIP2 in myeloid cells, such as monocytes and macrophages, other than the tunneling nanotube formation were not completely understood. Here, we provide evidence that TNFAIP2 functions as a unique regulator of CSF1R activation.

# Results

## TNFAIP2 inhibition or knockdown reduces CSF1R aggregate formation in macrophages

We initially analyzed the CSF-1–expanded self-renewing mouse bone marrow–derived macrophages (hereinafter referred to as CSF-1–expanded self-renewing BM-M$\Phi$) (34). These macrophages were established through the long-term culture of bone marrow cells in the presence of CSF-1 with repeated passages, and stably proliferated in the presence of CSF-1 (34). As observed in the mouse macrophage cell line BAC1.2F5 (22, 23), CSF1R was detected as large aggregates in the CSF-1–expanded self-renewing BM-M$\Phi$ (Fig 1A, upper). The CSF1R aggregates were detectable at the plasma membrane (Fig S1). Several CSF1R aggregates, but not all, localized to the Golgi (Fig S2). TNFAIP2

diffusely localized throughout the cytoplasm (Fig S3), but the large CSF1R aggregates were reduced by the TNFAIP2 inhibitor NPD3064 (TNFAIP2-i; Fig 1A, lower), the small chemical that inhibits TNFAIP2-mediated tunneling nanotube formation (31, 35, 36). The size of CSF1R aggregates of the TNFAIP2 inhibitor-treated cells was smaller than that of the vehicle (DMSO)-treated control cells (Fig 1B), although their cell surface expression level of CSF1R was comparable (Fig 1C). In contrast, there was no obvious effect of the TNFAIP2 inhibitor on the cellular distribution of the CSF-2 receptor $\alpha$ chain in the CSF-1–expanded self-renewing BM-M$\Phi$ (Fig S4).

We next compared the control and TNFAIP2 knockdown mouse macrophage cell line RAW264.7 cells that were established previously (26) (Fig S5). As observed for the CSF-1–expanded self-renewing BM-M$\Phi$, TNFAIP2 knockdown in RAW264.7 cells reduced the formation of large CSF1R aggregates (Fig 1D) and the size of the aggregates (Fig 1E), but did not affect the cell surface expression of CSF1R (Fig 1F).

## TNFAIP2 inhibition or knockdown reduces macrophage response to CSF-1, but not to CSF-2

We next examined how TNFAIP2 inhibition or knockdown affects the cellular response to CSF-1. The CSF-1–expanded self-renewing BM-M$\Phi$ proliferated in the presence of CSF-1, but not of CSF-2 (34), presumably because of their adaptation to CSF-1 during the long-term culture with CSF-1. However, they could survive even in the presence of CSF-2 alone (34). When added to the cultures containing CSF-1 or CSF-2, the TNFAIP2 inhibitor reduced the CSF-1–mediated proliferation from day 4 onward, but did not affect the CSF-2–mediated survival (Fig S6).

It is well known that CSF-2 or LPS up-regulates the expression of inducible nitric oxide synthase (iNOS) and that CSF-1 up-regulates the iNOS expression when combined with LPS in typical macrophages. In fact, CSF-1 up-regulated iNOS mRNA expression in the CSF-1–expanded self-renewing BM-M$\Phi$ when combined with 1 or 10 ng/ml LPS (Fig S7). However, unlike typical macrophages, CSF-1 alone was enough to up-regulate iNOS mRNA expression at a detectable level in the CSF-1–expanded self-renewing BM-M$\Phi$ (Fig S7), presumably because of their adaptation to CSF-1 during the long-term culture with CSF-1. Interestingly, the TNFAIP2 inhibitor reduced the CSF-1–mediated iNOS mRNA up-regulation, but did not affect the CSF-2–mediated iNOS mRNA up-regulation (Fig 2A).

The similar result was observed for RAW264.7 cells (Fig 2B): the TNFAIP2 knockdown reduced iNOS mRNA up-regulation by CSF-1 (left), but not that by CSF-2 (middle) or LPS (right). Consistent with this, the TNFAIP2 knockdown reduced the up-regulation of the production of nitric oxide by CSF-1, but not that by CSF2 or LPS (Fig 2C). The reduced response to CSF-1 by TNFAIP2 knockdown was also observed in the phagocytosis assay (Fig 2D). The TNFAIP2 knockdown cells showed a higher basal level of phagocytic activity than the control cells (Fig 2D, "0 h"), for unknown reasons. When added to these cells, CSF-1 enhanced the phagocytic activity in the control cells, but only slightly in the TNFAIP2 knockdown cells despite their higher basal phagocytic activity (Fig 2D, left). In sharp contrast, CSF-2 enhanced the phagocytic activity in both the control and TNFAIP2 knockdown cells regardless of their different levels of basal phagocytic activity (Fig 2D, right).

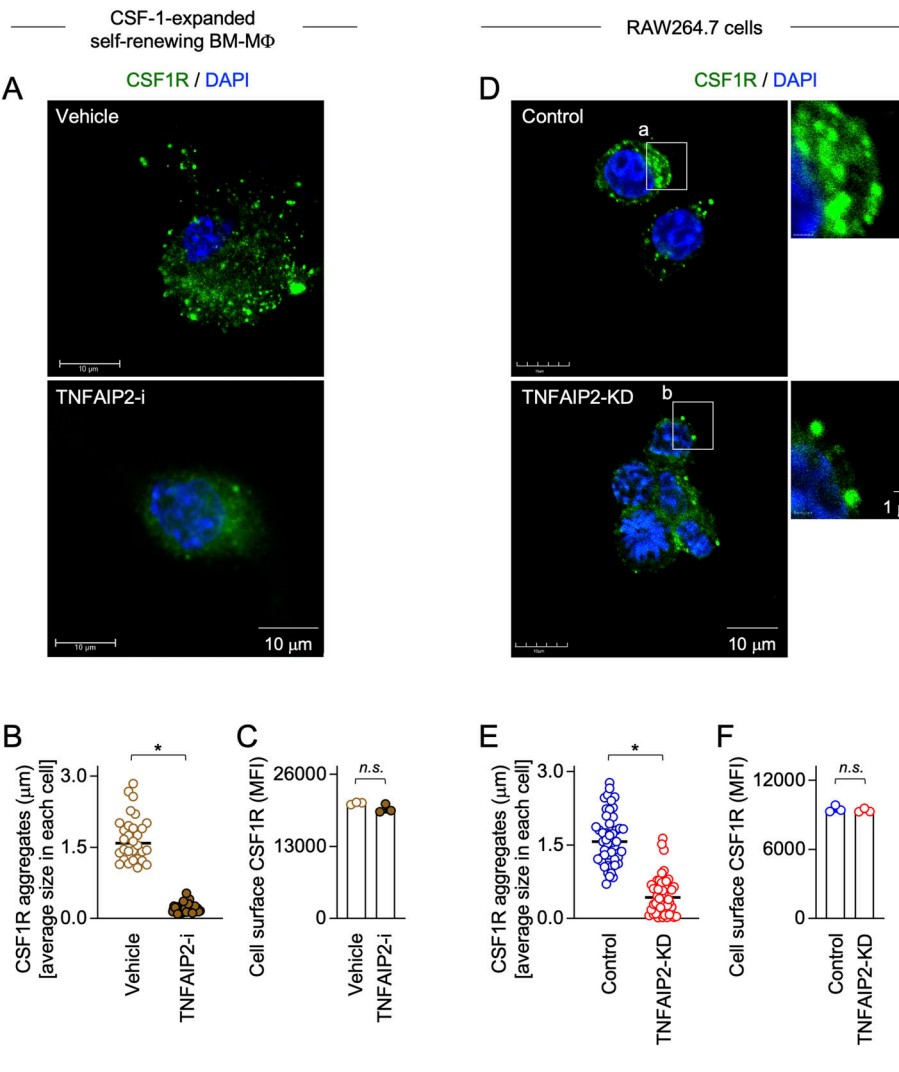

**Figure 1. Effect of TNFAIP2 inhibition or knockdown on CSF1R aggregate formation in macrophages.**
**(A)** CSF-1–expanded self-renewing BM-MΦ were maintained with 100 ng/ml CSF-1 and treated with DMSO (Vehicle) or 10 μM TNFAIP2 inhibitor NPD3064 (TNFAIP2-i) for 48 h in the presence of 100 ng/ml CSF-1. Then, the cells were CSF-1–starved for 6 h, co-stained with anti-CSF1R antibody (green) and DAPI (blue), and analyzed by immunofluorescence. Scale bar: 10 μm. **(A, B)** CSF-1–expanded self-renewing BM-MΦ were analyzed as in (A), and the average size of CSF1R aggregates in each cell is summarized (30 cells for each group). *$P < 0.05$. **(C)** CSF-1–expanded self-renewing BM-MΦ were maintained with 100 ng/ml CSF-1 and treated with DMSO (Vehicle) or 10 μM TNFAIP2 inhibitor NPD3064 (TNFAIP2-i) for 48 h in the presence of 100 ng/ml CSF-1. Then, the cells were CSF-1–starved for 6 h and analyzed for the cell surface expression of CSF1R by flow cytometry. The mean fluorescence intensity is shown. *n.s.*, not significant. **(D)** Control or TNFAIP2 knockdown (TNFAIP2-KD) RAW264.7 cells were co-stained with anti-CSF1R antibody (green) and DAPI (blue) and analyzed by immunofluorescence. Scale bar: 10 μm for left panels and 1 μm for right panels. **(D, E)** Control or TNFAIP2 knockdown (TNFAIP2-KD) RAW264.7 cells were analyzed as in (D), and the average size of CSF1R aggregates in each cell is summarized (50 cells for each group). *$P < 0.05$. **(F)** Control or TNFAIP2 knockdown (TNFAIP2-KD) RAW264.7 cells were analyzed for the cell surface expression of CSF1R by flow cytometry. The mean fluorescence intensity is shown. *n.s.*, not significant.

## TNFAIP2 knockdown reduces MAP kinase activation in macrophages, in response to CSF-1, but not to CSF-2

We next examined how TNFAIP2 knockdown affects signal activation in response to CSF-1. CSF-1 activated MAP kinases (ERK, p38, and JNK) in the control RAW264.7 cells, but weakly in the TNFAIP2 knockdown cells (Fig 3A). Such differential response was not observed for the CSF-2 treatment (Fig 3B). In fact, CSF-2 activated Stat5 in both the control and TNFAIP2 knockdown cells at the similar level (Fig 3B, p-Stat5 blot). These results were well consistent with the finding that the TNFAIP2 knockdown reduced the cellular response to CSF-1, but not to CSF-2 (see Figs 2 and S6).

The expression of human TNFAIP2 in the TNFAIP2 knockdown mouse RAW264.7 cells rescued their weak responses to CSF-1, including MAP kinase activation (Fig 3C), iNOS mRNA expression and nitric oxide production (Fig 3D), phagocytic activity (Fig 3E), and cell motility or osteoclastic differentiation (data not shown). Thus, the weak cellular response to CSF-1 caused by the TNFAIP2 knockdown in RAW264.7 cells was not due to off-target effects.

## TNFAIP2 inhibition or knockdown reduces CSF-1–induced CSF1R activation in macrophages

CSF-2 receptors consist of a unique α-chain and common β-chain (4), which are structurally unrelated to receptor tyrosine kinases including CSF1R. Because TNFAIP2 inhibition or knockdown reduced the cellular response to CSF-1, but not to CSF-2, we next examined whether the TNFAIP2 inhibition or knockdown affects CSF1R activation. The activated CSF1R tyrosine phosphorylates its downstream proteins (6, 17, 19, 20). In fact, CSF-1 induced the tyrosine-phosphorylated 150–300 kD proteins in the CSF-1–expanded self-renewing BM-MΦ, but such change was modest when the cells were pretreated with the TNFAIP2 inhibitor (Fig 4A). The similar result was observed when the control and TNFAIP2 knockdown RAW264.7 cells were compared (Fig 4B, p-Tyr blot), which was consistent with the weak MAP kinase activation in response to CSF-1 in the TNFAIP2 knockdown cells (see Fig 3A).

There were two CSF1R bands in unstimulated RAW264.7 cells, and the upper band decreased after CSF-1 stimulation in the control

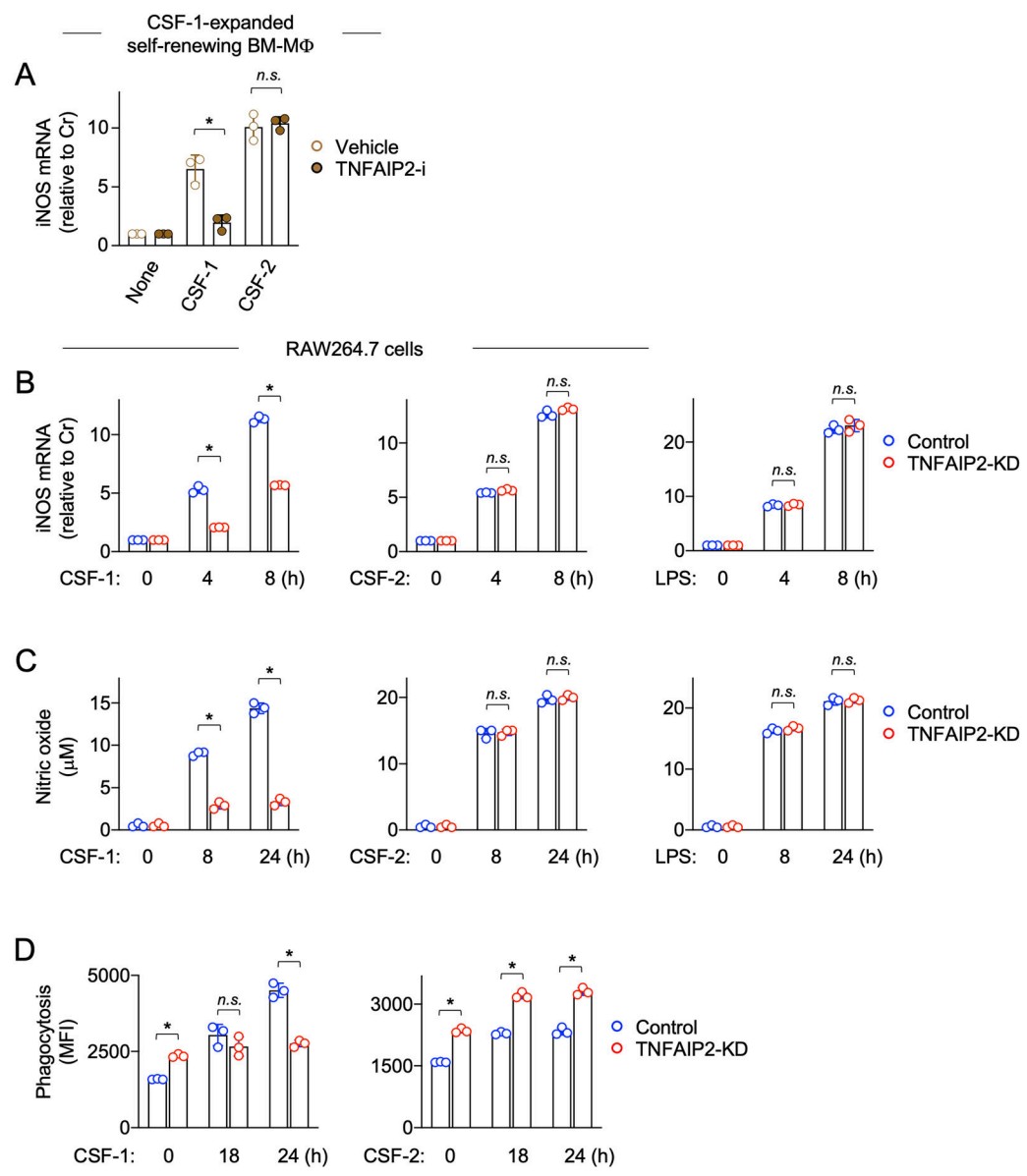

**Figure 2. Effect of TNFAIP2 inhibition or knockdown on the functional response of macrophages to CSF-1.**
**(A)** CSF-1–expanded self-renewing BM-MΦ were treated with DMSO (Vehicle) or 10 μM TNFAIP2 inhibitor NPD3064 (TNFAIP2-i) for 24 h in the presence of 100 ng/ml CSF-1. Then, the cells were left untreated (none) or treated with 100 ng/ml CSF-1 or 10 ng/ml CSF-2 for 8 h, and analyzed for iNOS mRNA expression by qRT–PCR (n = 3). The expression level shown is relative to that of untreated cells. *n.s.*, not significant. *$P < 0.05$. **(B)** Control or TNFAIP2 knockdown (TNFAIP2-KD) RAW264.7 cells were left untreated or treated with 100 ng/ml CSF-1, 10 ng/ml CSF-2, or 1 ng/ml LPS for 4 or 8 h, and analyzed for iNOS mRNA expression by qRT–PCR (n = 3). The expression level shown is relative to that of untreated cells. *n.s.*, not significant. *$P < 0.05$. **(C)** Control or TNFAIP2 knockdown (TNFAIP2-KD) RAW264.7 cells were left untreated or treated with 100 ng/ml CSF-1, 10 ng/ml CSF-2, or 1 ng/ml LPS for 8 or 24 h, and analyzed for the concentrations of nitric oxide in the supernatants using the Griess reagent (n = 3). *n.s.*, not significant. *$P < 0.05$. **(D)** Control or TNFAIP2 knockdown (TNFAIP2-KD) RAW264.7 cells were left untreated or treated with 100 ng/ml CSF-1 or 10 ng/ml CSF-2 for 18 or 24 h, and analyzed for phagocytic activity by flow cytometry (n = 3). The mean fluorescence intensity is shown. *n.s.*, not significant. *$P < 0.05$.

cells (Fig 4B, CSF1R blot/bar graph). This is because the upper band is the fully *N*-glycosylated cell surface form, whereas the lower is the hypo-*N*-glycosylated immature intracellular form (37), and the activation of the mature CSF1R is associated with its rapid poly-ubiquitination and apparent decrease in Western blotting (38). Importantly, such CSF1R change was modest in the TNFAIP2 knockdown cells (Fig 4B, CSF1R blot/bar graph). Consistent with this, the amount of tyrosine-phosphorylated CSF1R after CSF-1 stimulation, which reflects the activation and autophosphorylation of

CSF1R, was obvious in the control cells, but weak in the TNFAIP2 knockdown cells (Fig 4C).

### Exogenous TNFAIP2 expression augments CSF1R aggregate formation and CSF-1–mediated CSF1R activation in 293 cells

To further confirm that TNFAIP2 affects the CSF-1/CSF1R axis, we next performed the exogenous expression experiments using 293 cells, which are negative for the endogenous expression of TNFAIP2

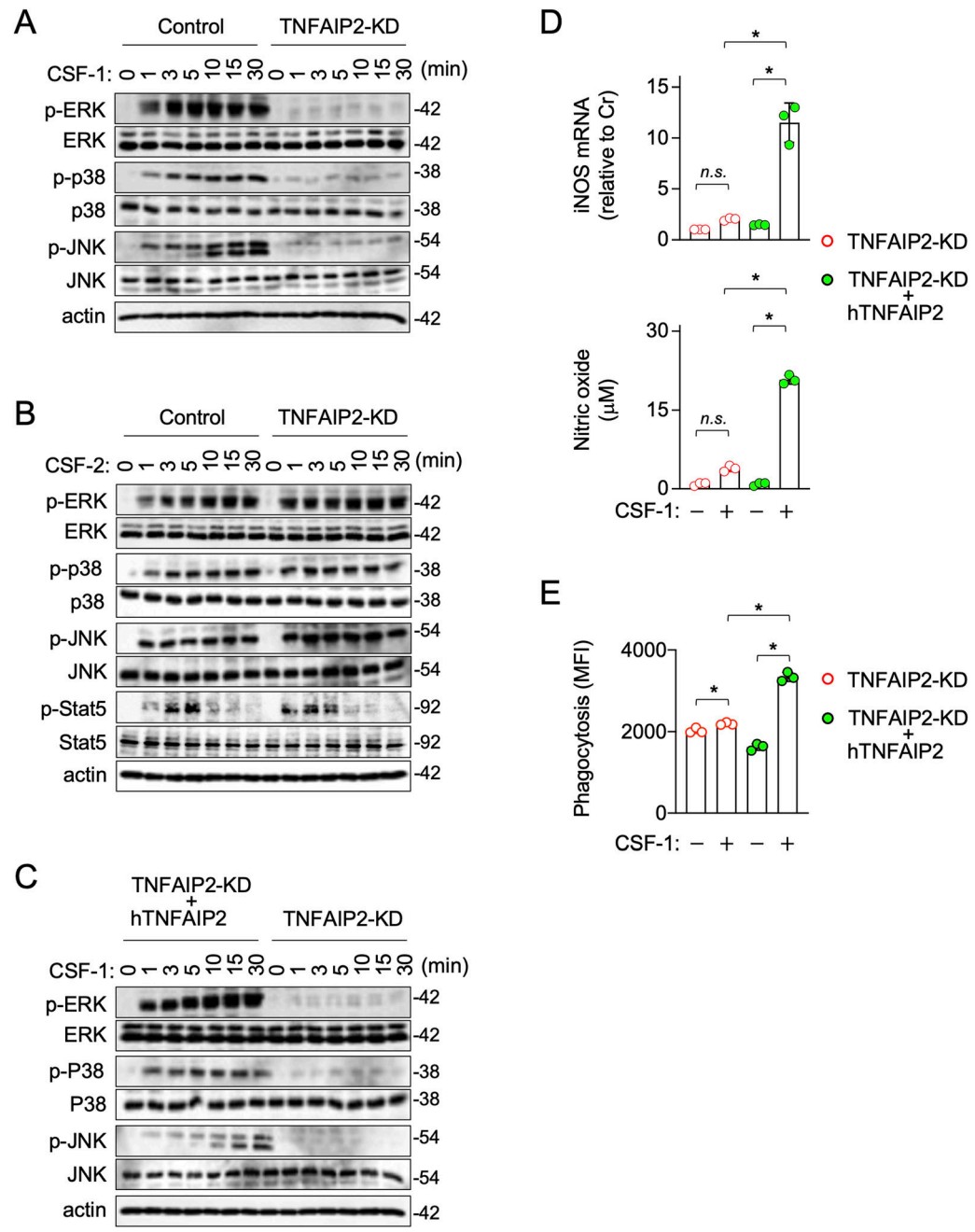

**Figure 3. Effect of TNFAIP2 knockdown on CSF-1–induced activation of MAP kinases in RAW264.7 cells, and effect of the exogenous expression of human TNFAIP2 on the response of TNFAIP2 knockdown RAW264.7 cells to CSF-1.**
**(A, B)** Control or TNFAIP2 knockdown (TNFAIP2-KD) RAW264.7 cells were serum-starved for 12 h, and left untreated or treated with 100 ng/ml CSF-1 (A) or 10 ng/ml CSF-2 (B) for the indicated periods. The total cell lysates were subjected to Western blotting. Antibodies used were as follows: anti-phosphorylated ERK (p-ERK), anti-total ERK, anti-phosphorylated p38 (p-p38), anti-total p38, anti-phosphorylated JNK (p-JNK), and anti-total JNK. In (B), the following antibodies were also used: anti-phosphorylated Stat5 (p-Stat5) and anti-total Stat5. The $\beta$-actin blot is the loading control. **(C)** TNFAIP2 knockdown (TNFAIP2-KD) or human TNFAIP2-expressing TNFAIP2 knockdown (TNFAIP2-KD + hTNFAIP2) RAW264.7 cells were serum-starved for 12 h, and left untreated or treated with 100 ng/ml CSF-1 for the indicated periods. **(A)** Total cell lysates were subjected to Western blotting, as in (A). **(D)** *Upper panel*, the TNFAIP2 knockdown (TNFAIP2-KD) or human TNFAIP2-expressing TNFAIP2 knockdown (TNFAIP2-KD + hTNFAIP2) RAW264.7 cells were left untreated or treated with 100 ng/ml CSF-1 for 8 h, and analyzed for iNOS mRNA expression by qRT–PCR (n = 3). The expression level shown is relative to that of untreated TNFAIP2-KD cells. *Lower panel*, the TNFAIP2 knockdown (TNFAIP2-KD) or human TNFAIP2-expressing TNFAIP2 knockdown (TNFAIP2-KD + hTNFAIP2) RAW264.7 cells were left untreated or treated with 100 ng/ml CSF-1 for 24 h, and analyzed for the concentrations of nitric oxide in the supernatants using the Griess reagent (n = 3). *n.s.*, not significant. $*P < 0.05$. **(E)** TNFAIP2 knockdown (TNFAIP2-KD) or human TNFAIP2-expressing TNFAIP2 knockdown (TNFAIP2-KD + hTNFAIP2) RAW264.7 cells were left untreated or treated with 100 ng/ml CSF-1 for 24 h, and analyzed for phagocytic activity by flow cytometry (n = 3). The mean fluorescence intensity is shown. $*P < 0.05$.

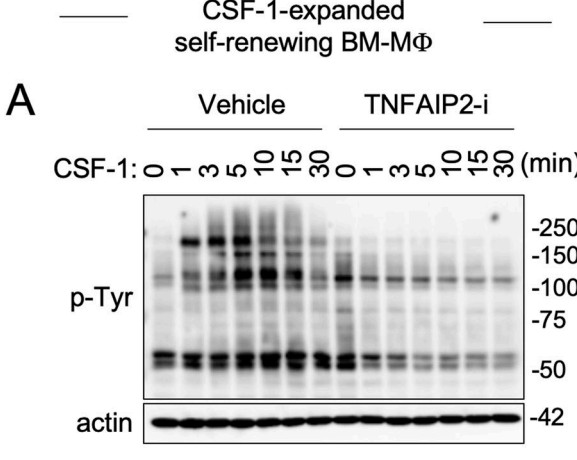

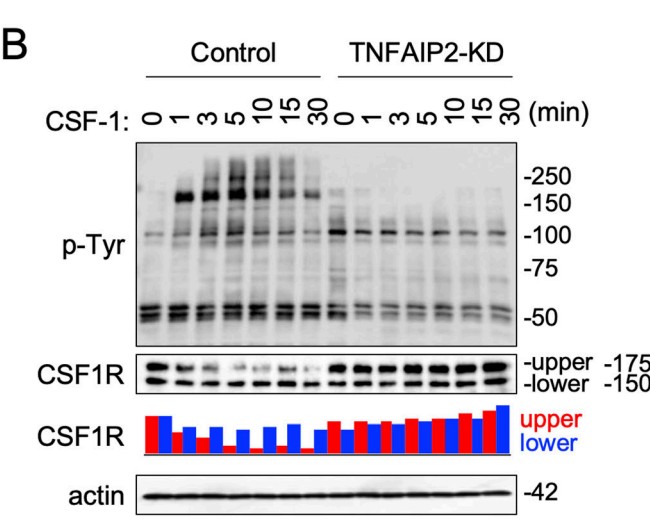

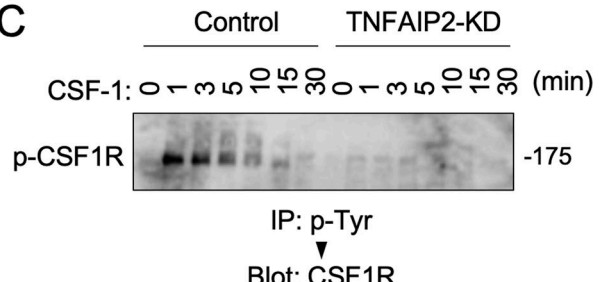

and CSF1R. The expression of TNFAIP2 in CSF1R-transfected 293 cells augmented the formation of large CSF1R aggregates (Fig 5A) and increased the size of CSF1R aggregates (Fig 5B). Furthermore, the TNFAIP2 expression augmented CSF-1–mediated activation of MAP kinases (Fig 5C) and phosphorylation of CSF1R at Tyr809 (Fig 5D), which is the major autophosphorylation site important for full activation of CSF1R (39). These results supported the idea that TNFAIP2 is involved in the large CSF1R aggregate formation and contributes to the efficient CSF-1–induced activation of CSF1R.

## CSF1R mutant lacking motif that mediates binding to phosphatidylinositol 4,5-bisphosphate (PIP2) fails to form large aggregates even in the presence of TNFAIP2

We next attempted to clarify how TNFAIP2 augments the CSF1R aggregate formation. Phosphatidylinositol 4,5-bisphosphate (PIP2) is the most abundant phosphoinositide and present in the inner leaflet of the plasma membrane (40, 41, 42), and TNFAIP2 is known to bind PIP2 via its N-terminal lysine-rich motifs (27). Interestingly, the similar lysine-rich motif is present in the intracellular juxtamembrane region of many receptor tyrosine kinases including CSF1R (43). The EGF receptor, the best characterized receptor tyrosine kinase, is supposed to bind PIP2 through such a lysine-rich motif (44). Thus, we next performed a series of experiments to test the hypothesis that PIP2 is required for TNFAIP2 to augment the CSF1R aggregate formation.

CSF1R has the lysine-rich putative PIP2-binding motif in its juxtamembrane region ($^{538}$YKYKQKPK$^{545}$, Fig 6A) (43). When transfected in TNFAIP2-expressing 293 cells, the 1–545 mutant, which lacked the most cytoplasmic domain but retained the $^{538}$YKYKQKPK$^{545}$ sequence, formed large aggregates (Fig 6A, upper), as the WT CSF1R did (see Fig 5A, lower). In contrast, the 1–537 mutant, which completely lacked the intracellular domain including the $^{538}$YKYKQKPK$^{545}$ sequence, failed to form such large aggregates (Fig 6B, lower). This difference between these mutants was confirmed by the quantification of the size of CSF1R aggregates (Fig 6C). The 1–537 mutant was expressed on the surface of the parent 293 cells, the level of which was higher than that of the WT CSF1R or comparable to that of the 1–545 mutant (Fig S8A, upper panels). Interestingly, the TNFAIP2 expression increased the cell surface expression level of the WT and the 1–545 mutant (Fig S8A, lower panels), which was consistent with their decreased intracellular level (Fig S8B). Such changes were not observed for the 1–537 mutant (Fig S8). These results suggest that the formation of CSF1R aggregates in the presence of TNFAIP2 is associated with the intracellular-to-cell surface distribution of CSF1R, and raise the possibility that the putative PIP2-binding motif in the juxtamembrane region of CSF1R is involved in the processes.

**Figure 4. Effect of TNFAIP2 inhibition or knockdown on CSF-1–induced CSF1R activation in macrophages.**
**(A)** CSF-1–expanded self-renewing BM-MΦ were pretreated with DMSO (Vehicle) or 10 μM TNFAIP2 inhibitor NPD3064 (TNFAIP2-i) overnight in the presence of 100 ng/ml CSF-1. Then, the cells were CSF-1/serum-starved for 6 h, and left untreated or treated with 100 ng/ml CSF-1 for the indicated periods. The total cell lysates were subjected to Western blotting using anti-phosphotyrosine (p-Tyr) antibody. The β-actin blot is the loading control. **(B)** Control or TNFAIP2 knockdown (TNFAIP2-KD) RAW264.7 cells were serum-starved for 12 h, and left untreated or treated with 100 ng/ml CSF-1 for the indicated periods. The total cell

lysates were subjected to Western blotting using anti-phosphotyrosine (p-Tyr) or anti-CSF1R antibody. The β-actin blot is the loading control. In the bar graph, the density of the upper (red) or lower (blue) band of CSF1R is summarized (see the text for details of the upper and lower CSF1R bands). The level shown is relative to that of untreated cells. **(C)** Control or TNFAIP2 knockdown (TNFAIP2-KD) RAW264.7 cells were serum-starved for 12 h, and left untreated or treated with 100 ng/ml CSF-1 for the indicated periods. The anti-phosphotyrosine immunoprecipitates (IP: p-Tyr) were analyzed by Western blotting using anti-CSF1R antibody (Blot: CSF1R), to detect phosphorylated CSF1R (p-CSF1R).

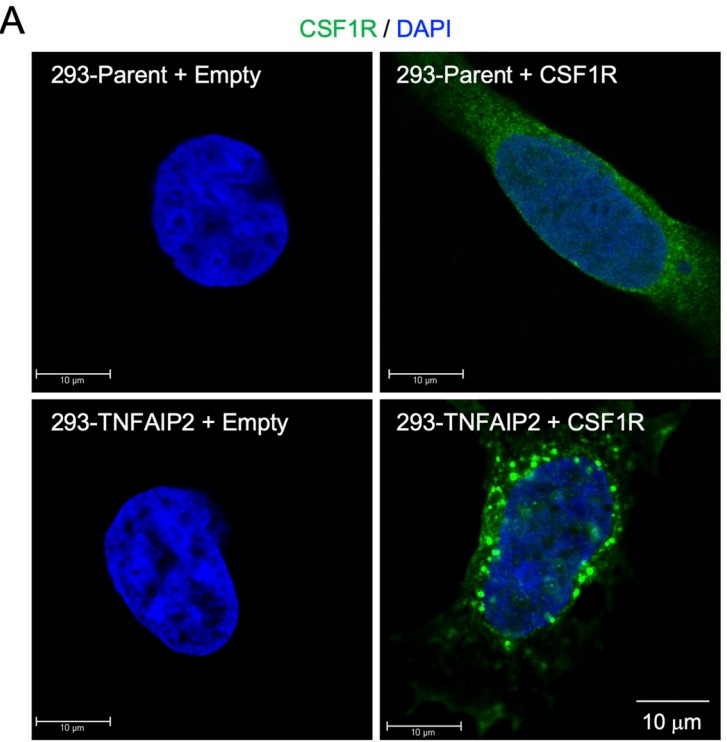

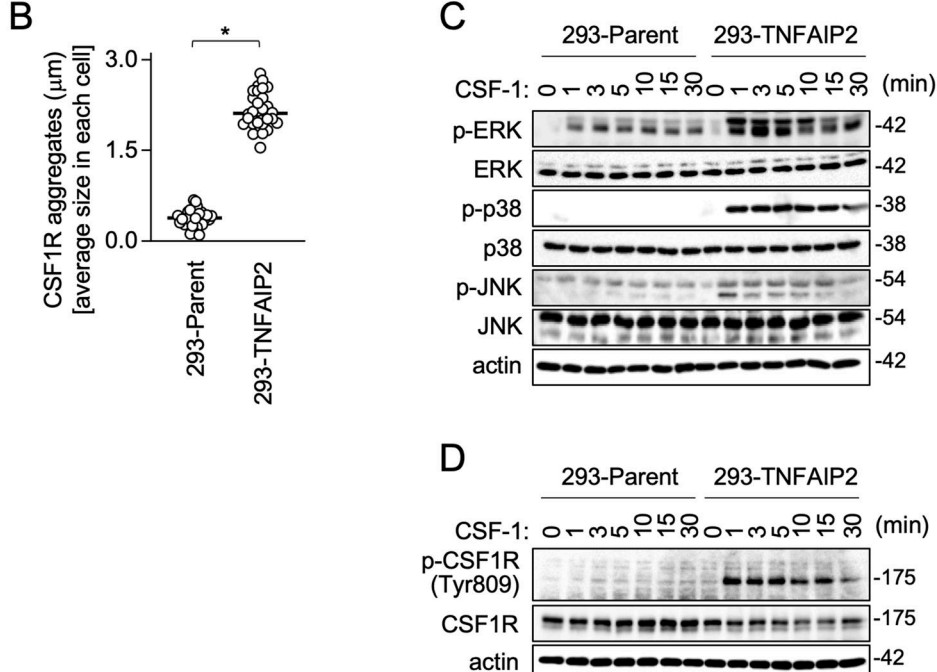

**Figure 5. Effect of TNFAIP2 expression on CSF1R aggregate formation and CSF-1–induced CSF1R activation in CSF1R-transfected 293 cells.**
**(A, B)** Parent or TNFAIP2-expressing 293 cells were transfected with the empty plasmid or the WT CSF1R plasmid as indicated, cultured for 2 d in the absence of CSF-1, co-stained with anti-CSF1R antibody (green) and DAPI (blue), and analyzed by immunofluorescence. Scale bar: 10 $\mu$m. In (B), the average size of CSF1R aggregates in each CSF1R-transfected cell is summarized (30 cells for each group). *$P < 0.05$. **(C)** Parent or TNFAIP2-expressing 293 cells were transfected with the WT CSF1R plasmid and cultured for 2 d in the absence of CSF-1. Then, they were serum-starved for 12 h, and left untreated or treated with 100 ng/ml CSF-1 for the indicated periods. The total cell lysates were subjected to Western blotting. Antibodies used were as follows: anti-phosphorylated ERK (p-ERK), anti-total ERK, anti-phosphorylated p38 (p-p38), anti-total p38, anti-phosphorylated JNK (p-JNK), and anti-total JNK. The $\beta$-actin blot is the loading control. **(C, D)** Total cell lysates of the CSF1R-transfected parent 293 cells or CSF1R-transfected TNFAIP2-expressing 293 cells were prepared as in (C) and analyzed by Western blotting using the antibody specific for tyrosine-809-phosphorylated CSF1R (p-CSF1R [Tyr809]). Total CSF1R was also analyzed as a reference. The $\beta$-actin blot is the loading control.

## TNFAIP2 mutant lacking the PIP2-binding motif fails to augment large CSF1R aggregate formation

TNFAIP2 has two lysine-rich motifs at its N terminus (see Fig 7A), and the mutant lacking the first motif (ΔK1) or the mutant lacking the second motif (ΔK2) lost the binding to PIP2 (27). Thus, we

established 293 cells expressing the TNFAIP2 ΔK1 or ΔK2 mutant (Fig 7B). When transfected with CSF1R, these cells expressed CSF1R, the level of which was comparable to that of 293 cells expressing the WT TNFAIP2 (Fig 7C). However, the size of CSF1R aggregates in these cells was markedly smaller than that of cells expressing the WT TNFAIP2 (Fig 7D). The result further suggests a possible

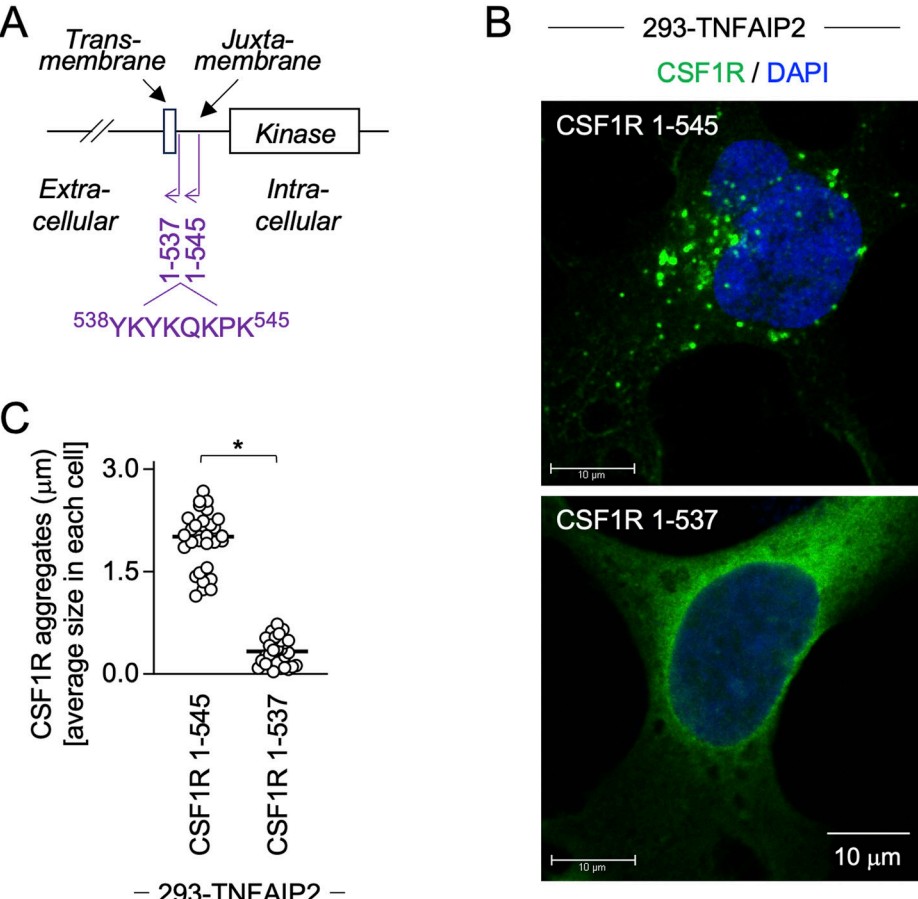

**Figure 6. Aggregate formation of CSF1R mutants in TNFAIP2-expressing 293 cells.**
**(A)** C-terminally deleted CSF1R mutants (1–537 and 1–545) are schematically shown. The amino acid sequence ($^{538}$YKYKQKPK$^{545}$) of the lysine-rich putative PIP2-binding motif is also shown. **(B, C)** TNFAIP2-expressing 293 cells were transfected with the 1–545 or the 1–537 mutant CSF1R plasmid, cultured for 2 d in the absence of CSF-1, co-stained with anti-CSF1R antibody (green) and DAPI (blue), and analyzed by immunofluorescence. Scale bar: 10 $\mu$m. In (C), the average size of CSF1R aggregates in each cell is summarized (30 cells for each group). *$P < 0.05$.

---

involvement of PIP2 in the TNFAIP2-mediated large CSF1R aggregate formation.

### Depletion of cellular PIP2 reduces TNFAIP2-mediated large CSF1R aggregate formation

To further confirm the involvement of PIP2 in the TNFAIP2-mediated large CSF1R aggregate formation, we performed the experiment using the plasmid expressing inositol polyphosphate 5-phosphatase type IV (5ptase), which reduces the cellular level of PIP2 (45, 46). When expressed in 293 cells expressing TNFAIP2, 5ptase did not affect CSF1R expression (Fig 8A), but reduced the formation of large CSF1R aggregates (Fig 8B) and the size of CSF1R aggregates (Fig 8C). These results, together with the results of experiments using CSF1R mutants (see Fig 6) and TNFAIP2 mutants (see Fig 7), strongly suggest that PIP2 is required for TNFAIP2 to augment the CSF1R aggregate formation.

### TNFAIP2 alters the cellular distribution of PIP2

We next performed the experiment using the plasmid expressing the GFP-fused pleckstrin homology domain of phospholipase C$\delta$ (GFP-PLC$\delta$-PH), which is widely used as a probe to visualize cellular PIP2 distribution in living cells (46, 47, 48). Interestingly, the parent 293 cells

and TNFAIP2-expressing 293 cells showed a marked difference in the distribution of GFP-PLC$\delta$-PH: the large GFP-PLC$\delta$-PH–positive structures were observed in the TNFAIP2-expressing 293 cells, but not in the parent 293 cells (Fig 9A). Such large GFP-PLC$\delta$-PH–positive structures were also observed in 293 cells co-expressing TNFAIP2 and CSF1R, but not in 293 cells expressing CSF1R alone (Fig S9A). Because there was no difference in the percentage of GFP-PLC$\delta$-PH–positive cells or the expression level of GFP-PLC$\delta$-PH between TNFAIP2-non-expressing 293 cells and TNFAIP2-expressing 293 cells (Figs 9B and S9B), these results suggest that TNFAIP2 alters the cellular distribution of PIP2, which leads to the formation of large CSF1R aggregates.

### Human peripheral blood monocyte–derived macrophages form CSF1R aggregates, and TNFAIP2 inhibition reduces CSF-1–mediated survival of monocyte-derived macrophages

Finally, we performed the experiments using human peripheral blood monocytes and monocyte-derived macrophages, the latter of which were prepared by culturing monocytes for 5 d in the presence of CSF-1 (49). The CSF1R aggregates were hardly detected in monocytes (Fig 10A), but readily observed in monocyte-derived macrophages (Fig 10B), which was presumably due to a higher expression of both CSF1R and TNFAIP2 in monocyte-derived macrophages (Fig S10). Furthermore, the TNFAIP2 inhibitor did not affect

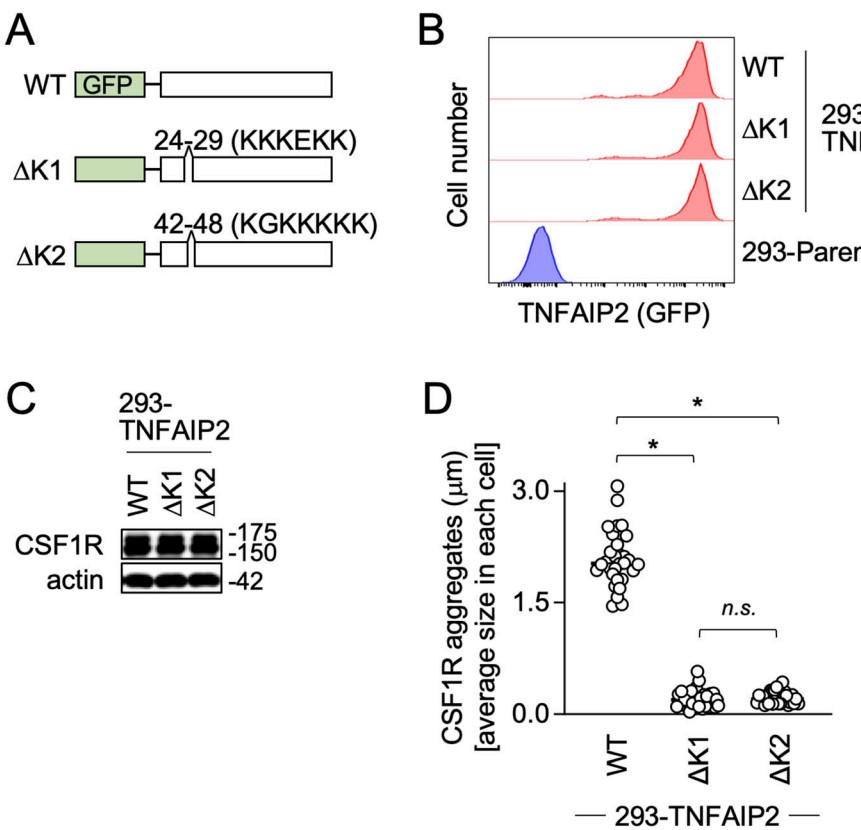

**Figure 7.   CSF1R aggregate formation in mutant TNFAIP2-expressing CSF1R-transfected 293 cells.**
**(A)** GFP-fused WT TNFAIP2 and its mutants (ΔK1 and ΔK2) used are schematically shown. The ΔK1 and ΔK2 lack the first ($^{24}$KKKEKK$^{29}$) and second ($^{42}$KGKKKKK$^{48}$) lysine-rich PIP2-binding motif, respectively. **(B)** 293 cells were engineered to stably express TNFAIP2 WT, ΔK1, or ΔK2. The expression of these GFP-fused TNFAIP2 proteins was confirmed by flow cytometry. The cells that were not transfected with the CSF1R plasmid were analyzed. The parent 293 cells were also analyzed as a reference. **(C)** 293 cells expressing TNFAIP2 WT, ΔK1, or ΔK2 were transfected with the CSF1R plasmid, cultured for 2 d in the absence of CSF-1, and analyzed for the expression level of CSF1R by Western blotting. The β-actin blot is the loading control. **(D)** 293 cells expressing TNFAIP2 WT, ΔK1, or ΔK2 were transfected with the CSF1R plasmid, cultured for 2 d in the absence of CSF-1, stained with anti-CSF1R antibody, and analyzed by immunofluorescence. Then, the average size of CSF1R aggregates in each cell is summarized (30 cells for each group). n.s., not significant. *$P < 0.05$.

the CSF-1–mediated survival of monocytes (Fig 10C), but reduced the CSF-1–mediated survival of monocyte-derived macrophages (Fig 10D, upper). Such reduction was not observed for the CSF-2–mediated survival of monocyte-derived macrophages (Fig 10D, lower). These results suggest that TNFAIP2 functions as the regulator of CSF1R in macrophages, but not in undifferentiated monocytes. This idea is further supported by the experiments using the CSF-1–expanded self-renewing mouse bone marrow–derived macrophages and the mouse macrophage cell line RAW264.7 cells.

## Discussion

In this study, we identified an unreported function of TNFAIP2 in macrophages. We revealed that TNFAIP2 is involved in the efficient functional and signaling response of macrophages to CSF-1, but not to CSF-2. In fact, TNFAIP2 contributes to the efficient CSF-1–induced activation of its receptor CSF1R. We also revealed that TNFAIP2 augments for the formation of large CSF1R aggregates, which have been supposed to be beneficial for CSF-1 to dimerize and activate CSF1R (21). The cellular phosphoinositide PIP2 is required for the formation of the large CSF1R aggregates mediated by TNFAIP2. Collectively, it is likely that TNFAIP2 regulates the cellular localization of CSF1R via PIP2 and augments CSF-1–induced activation of CSF1R. Thus, TNFAIP2 appears a unique cellular regulator of CSF1R activation.

In the absence of CSF-1 or IL-34, CSF1R is present as the inactive monomer because of cis-autoinhibition (16, 17). The binding of CSF-1 or IL-34, both of which are homodimers, induces the dimerization of CSF1R and releases the cis-autoinhibition (16, 17, 18). Thus, the dimerization is the critical step for the activation of CSF1R, and the abundance or distribution of CSF1R at the plasma membrane is the rate-limiting step in its activation. Because of this, the large CSF1R aggregates in which the monomers are close to each other are supposed to be beneficial for CSF-1 or IL-34 to dimerize CSF1R (21). Our findings suggest that TNFAIP2 brings the CSF1R monomers close to each other via PIP2 and therefore facilitates the efficient dimerization and activation of CSF1R in macrophages, in response to CSF-1 or IL-34.

The present study at least in part answers the two long-standing questions, namely, the mechanisms by which CSF1R forms large aggregates in macrophages and the significance of the large CSF1R aggregates for CSF-1–induced CSF1R activation. Apart from its physiologically high expression in macrophages, TNFAIP2 is aberrantly expressed in nasopharyngeal carcinoma cells (28). Interestingly, Liu and Wu recently reported that TNFAIP2 knockdown in nasopharyngeal carcinoma cells reduced the autophosphorylation of the receptor tyrosine kinase EGF receptor in response to its ligand EGF ((50); meeting abstract). Although the details including underlying mechanisms are unclear, it is possible that the reduced cellular response to EGF by TNFAIP2 knockdown is due to altered cellular localization of the EGF receptor, as exemplified by CSF1R. In fact, the EGF receptor is supposed to bind PIP2 through its lysine-rich motif (44). Thus, TNFAIP2 may regulate not only CSF1R but also other receptor tyrosine kinases including the EGF receptor.

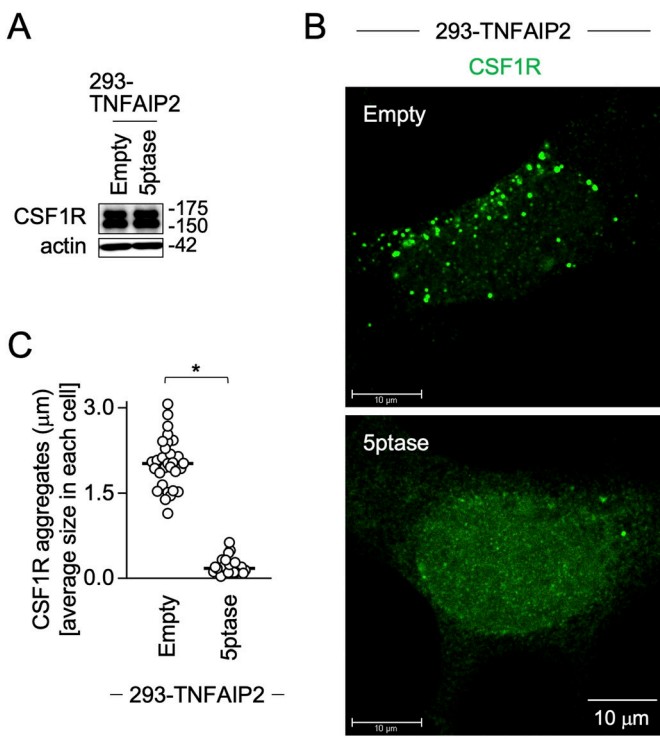

**Figure 8. Effect of inositol polyphosphate 5-phosphatase type IV expression on TNFAIP2-mediated CSF1R aggregate formation.**
**(A)** TNFAIP2-expressing 293 cells were co-transfected with the CSF1R plasmid and either the empty plasmid (Empty) or inositol polyphosphate 5-phosphatase type IV plasmid (5ptase). Then, the cells were cultured for 2 d in the absence of CSF-1 and analyzed for the expression level of CSF1R by Western blotting. The β-actin blot is the loading control. **(B)** TNFAIP2-expressing 293 cells were co-transfected with the CSF1R plasmid and either the empty (Empty) or 5ptase plasmid (5ptase). Then, the cells were cultured for 2 d in the absence of CSF-1, stained with anti-CSF1R antibody (green) and analyzed by immunofluorescence. Scale bar: 10 μm. **(B, C)** TNFAIP2-expressing 293 cells were analyzed as in (B), and the average size of CSF1R aggregates in each cell is summarized (30 cells for each group). *$P < 0.05$.

HTLV-1 is the causative virus of adult T-cell leukemia. We previously reported that HTLV-1 induces the aberrant expression of TNFAIP2 in CD4+ T cells and that TNFAIP2 inhibition or knockdown alters the cellular localization of Gag, the structural protein of HTLV-1 (33). Because Gag has a lysine- and arginine-rich polybasic motif at its N terminus (51), it is possible that the altered Gag localization by the TNFAIP2 inhibition/knockdown is related to the binding of Gag to PIP2 through the motif.

Our results suggest that TNFAIP2 alters the cellular distribution of PIP2, which leads to the formation of large CSF1R aggregates. However, how TNFAIP2 alters the cellular distribution of PIP2 is unclear. TNFAIP2 is thought to initiate the formation of tunneling nanotubes through RalA, a small GTPase, and the exocyst complex as the downstream effector of RalA (26, 27), although the precise role of TNFAIP2, RalA, and the exocyst complex is not fully understood. The exocyst complex is evolutionarily conserved and known to tether secretory vesicles to the plasma membrane (52, 53). Among eight components of the exocyst complex, Sec3 (also known as EXOC1) and Exo70 (also known as EXOC7) bind PIP2 (54, 55). Thus, these components may help TNFAIP2 to alter cellular PIP2 distribution.

A number of cellular proteins have been proposed to bind PIP2 and induce the clustering of PIP2 (41). However, the binding and clustering may not be enough to explain the altered cellular PIP2 distribution induced by TNFAIP2 because the PIP2 signals in the presence of TNFAIP2 were very large (see Fig 9). The formation of tunneling nanotubes, the hallmark function of TNFAIP2, may help to induce such large PIP2 accumulation.

Jin et al recently reported that TNFAIP2 knockdown in macrophages reduced the activation of MAP kinases in response to $H_2O_2$ (56). However, the underlying mechanisms are unknown. Interestingly, $H_2O_2$ activates a $Ca^{2+}$-permeable channel TRPM2 (formerly known as LTRPC2) (57), and PIP2 was proposed to regulate the activation of TRPM2 (58). Thus, the PIP2 may be involved in the reduced activation of MAP kinases in the TNFAIP2 knockdown macrophages.

It is possible that the formation of CSF1R aggregates by TNFAIP2 is due to their interaction. In our co-immunoprecipitation using TNFAIP2-expressing CSF1R-transfected 293 cells, the WT CSF1R and the 1–545 mutant, but not the 1–537 mutant, were detected in the anti-TNFAIP2 immunoprecipitates (Fig S11), which was consistent with the finding that the WT CSF1R and the 1–545 mutant, but not the 1–537 mutant, formed aggregates in the presence of TNFAIP2. However, it should be mentioned that TNFAIP2 does not necessarily co-localize with CSF1R because it diffusely localizes throughout the cytoplasm in the transfected 293 cells (Fig S12). The diffuse localization of TNFAIP2 does not preclude the importance of its interaction with CSF1R, but further studies are necessary to understand to what degree the possible TNFAIP2/CSF1R interaction contributes to the formation of CSF1R aggregates by TNFAIP2. It is also necessary to clarify whether TNFAIP2 directly or indirectly interacts with CSF1R.

In this study, we used CSF-1 as the ligand of CSF1R and have demonstrated that TNFAIP2 regulates the CSF-1/CSF1R axis. Because IL-34 is the alternative ligand of CSF1R, TNFAIP2 may also regulate the IL-34/CSF1R axis. However, more studies are necessary to clarify whether the formation of large CSF1R aggregates via PIP2 is the sole mechanism by which TNFAIP2 augments the functional and signaling response of macrophages to CSF-1, because PIP2 is known to support a broad spectrum of cellular functions and the best characterized regulator of actin cytoskeleton among all phosphoinositides (40, 41, 42). The present study suggests that TNFAIP2 functions as the regulator of CSF1R in macrophages, but not in undifferentiated monocytes. Therefore, it will be interesting to analyze how monocytes and tissue-resident macrophages are differently affected by the TNFAIP2 knockout in mice. Despite the unresolved questions, TNFAIP2 will be helpful to understand the mechanism of activation of CSF1R in macrophages and other receptor tyrosine kinases including the EGF receptor.

# Materials and Methods

### CSF-1–expanded self-renewing BM-MΦ

Mouse bone marrow–derived macrophages that stably proliferated in the presence of CSF-1 were established previously (34). They were

## A

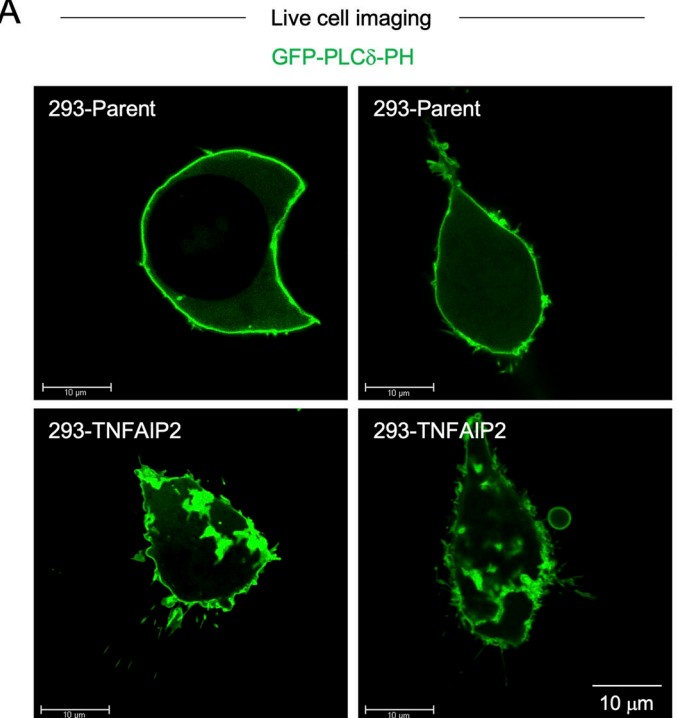

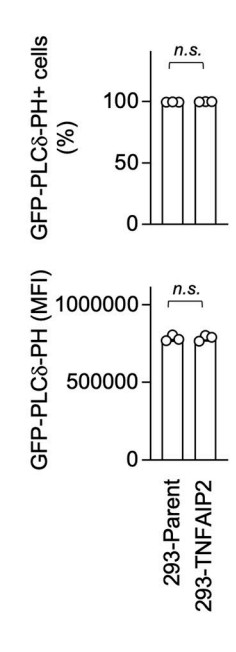

**Figure 9. Effect of TNFAIP2 on the cellular distribution of PIP2.**

**(A, B)** Parent or TNFAIP2-expressing 293 cells were transfected with the GFP-PLCδ-PH plasmid and cultured for 2 d. The cells that were not transfected with the CSF1R plasmid were analyzed. In (A), the cells were analyzed for GFP expression by live-cell imaging to visualize the cellular distribution of PIP2. Two images are shown for each group. Scale bar: 10 μm. In (B), the cells were analyzed for GFP expression (to detect GFP-PLCδ-PH) by flow cytometry (n = 3). The percentage of GFP-positive cells and mean fluorescence intensity are summarized in the upper and lower panels, respectively. *n.s.*, not significant.

maintained with RPMI 1640 medium/10% FCS containing recombinant human (rh) CSF-1 at a final concentration of 100 ng/ml.

### RAW264.7 cells

The control and TNFAIP2 knockdown mouse macrophage–like RAW264.7 cells were established previously (26). They were maintained with RPMI 1640 medium/10% FCS. The control or TNFAIP2 knockdown cells were further engineered to express human TNFAIP2. The N-terminally Flag-tagged human TNFAIP2 cDNA (NCBI GenBank M92357.1) was synthesized at Eurofins Japan and cloned into the lentiviral vector pLVSIN-EF1α-Hyg (TaKaRa-Bio). The lentivirus was prepared and added to the control or TNFAIP2 knockdown cells, and cells expressing human TNFAIP2 were selected under the cultures containing 750 μg/ml hygromycin B.

### Human peripheral blood monocytes and monocyte-derived macrophages

The approval for this study was obtained from medical ethical committees of Kumamoto University, Japan. PBMCs were collected from healthy volunteers after informed consent had been obtained in accordance with the Declaration of Helsinki. Monocytes were sorted from PBMCs, using anti-CD14 magnetic particles (clone MΦP9; BD Biosciences) and the cell separation magnet (IMag; BD Biosciences) (49). Monocyte-derived macrophages were also prepared. In brief, the sorted monocytes were differentiated into macrophages by culturing with RPMI 1640 medium/10% FCS containing 100 ng/ml rhCSF-1 for 5 d (49).

### CSF-1 and CSF-2

rhCSF-1 was a kind gift from the Morinaga Milk Industry and used at a final concentration of 100 ng/ml. The recombinant mouse (rm) CSF-2 (Miltenyi Biotec) was used at a final concentration of 10 ng/ml. In selected experiments, the recombinant human (rh) CSF-2 (Miltenyi Biotec) was used at a final concentration of 10 ng/ml.

### LPS

LPS of *Escherichia coli* serotype 0111:B4 was purchased from Alexis and added to the culture at a final concentration of 1 or 10 ng/ml.

### TNFAIP2 inhibitor

The TNFAIP2 inhibitor NPD3064 was identified by an affinity-based chemical array screening (31). In brief, test compounds were arrayed onto the photoaffinity linker–coated slides, which were incubated with the lysates of 293 cells expressing either the control DsRed or the DsRed-TNFAIP2 fusion protein. The fluorescent signals were quantified, and the compound NPD3064 to which DsRed-TNFAIP2, but not DsRed, bound was identified. The TNFAIP2 inhibitor was synthesized at Pharmeks, dissolved in DMSO, and added to cultures at a final concentration of 5, 10, or 20 μM (0.1% vol/vol). The same volume of DMSO was added as a vehicle control.

### 293 cells stably expressing TNFAIP2

The 293A cells (Invitrogen) were maintained with DMEM /10% FCS and engineered to express human TNFAIP2 using the lentiviral

## Human monocytes

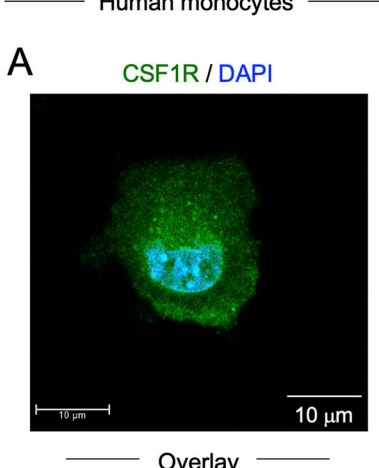

**A** CSF1R / DAPI

10 µm

Overlay

## Monocyte-derived MΦ

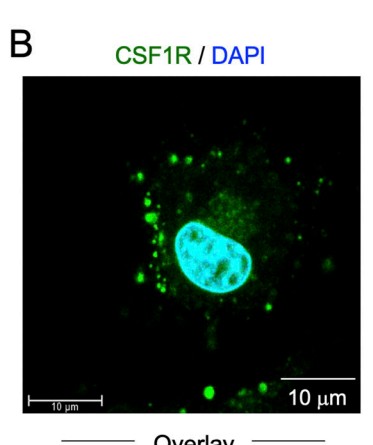

**B** CSF1R / DAPI

10 µm

Overlay

**Figure 10. Localization of CSF1R in human monocytes or monocyte-derived macrophages, and effect of TNFAIP2 inhibitor on their survival.**

**(A)** Human peripheral blood CD14⁺ monocytes were freshly prepared, co-stained with anti-CSF1R antibody (green) and DAPI (blue), and analyzed by immunofluorescence. The overlay images composed of serial Z-sections are shown. Scale bar: 10 µm. **(B)** Macrophages were prepared by culturing human peripheral blood CD14⁺ monocytes in the presence of 100 ng/ml CSF-1 for 5 d. The monocyte-derived macrophages were CSF-1–starved for 6 h, co-stained with anti-CSF1R antibody (green) and DAPI (blue), and analyzed by immunofluorescence. The overlay images composed of serial Z-sections are shown. Scale bar: 10 µm. **(C)** Human peripheral blood CD14⁺ monocytes obtained from three different donors were seeded in the presence of 100 ng/ml CSF-1 (upper) or 10 ng/ml CSF-2 (lower) and treated with DMSO (Vehicle), or 10 or 20 µM TNFAIP2 inhibitor NDP3064 (TNFAIP2-i). The media were replaced with the fresh complete media containing the cytokine (CSF-1 or CSF-2) and the inhibitor (DMSO or TNFAIP2-i) at day 3. The cell survival mediated by CSF-1 (upper) or CSF-2 (lower) was monitored at day 5, by the MTT assay (n = 3). *n.s.*, not significant. **(B, D)** Monocyte-derived macrophages prepared from three different donors as in (B). They were cultured in the presence of 100 ng/ml CSF-1 (upper) or 10 ng/ml CSF-2 (lower) and treated with DMSO (Vehicle), or 10 or 20 µM TNFAIP2 inhibitor NDP3064 (TNFAIP2-i). The media were replaced with the fresh complete media containing the cytokine (CSF-1 or CSF-2) and the inhibitor (DMSO or TNFAIP2-i) at day 3. The cell survival mediated by CSF-1 (upper) or CSF-2 (lower) was monitored at day 5, by the MTT assay (n = 3). *n.s.*, not significant. *$P < 0.05$.

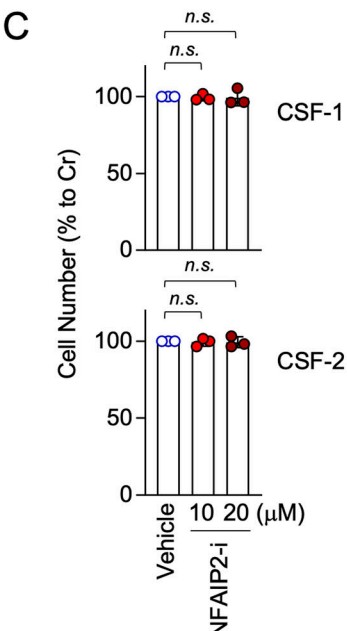

**C**

*n.s.* *n.s.*

CSF-1

*n.s.* *n.s.*

CSF-2

Vehicle — 10 20 (µM) — TNFAIP2-i

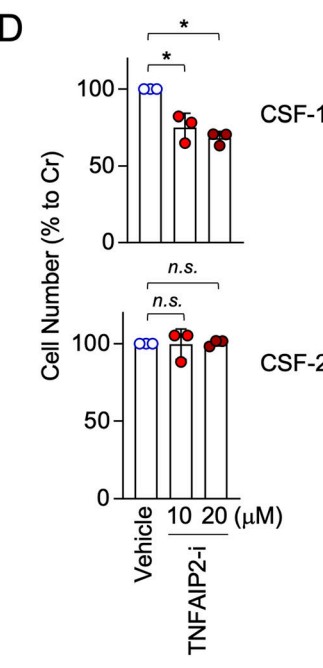

**D**

* *

CSF-1

*n.s.* *n.s.*

CSF-2

Vehicle — 10 20 (µM) — TNFAIP2-i

system. The lentivirus was added to 293 cells, and cells expressing human TNFAIP2 were selected under the cultures containing 200 µg/ml hygromycin B. For selected experiments, the parent or TNFAIP2-expressing 293 cells were further engineered to express human CSF1R. They were transfected with the CSF1R plasmid (37), using Lipofectamine 3000 reagent (Invitrogen). The cells expressing CSF1R were selected under the cultures containing 1,200 µg/ml G418 and enriched by cell sorting using FACSAria II (BD Biosciences). The 293A cells were also engineered to express GFP-fused mouse TNFAIP2. They were transfected with the plasmid encoding the WT or mutant (ΔK1 or ΔK2) (27), using Lipofectamine 3000 reagent. The cells expressing GFP-fused TNFAIP2 were selected under the cultures containing 1,200 µg/ml G418 and enriched by cell sorting using FACSAria II.

## Transfection using 293 cells

The 293 cells were seeded onto 12-well plates, cultured, and transfected with various plasmids using 3 µl Lipofectamine 3000 reagent and 2 µl P3000 reagent (both from Invitrogen). The total amount (1 µg) of the plasmid was normalized using appropriate control (Empty) vectors. After 6 h of transfection, the culture medium was replaced with fresh medium. The cells were further cultured for 1 or 2 d and subjected to immunofluorescence, live-cell imaging, flow cytometry, or Western blotting.

## Plasmids

The C-terminally deleted CSF1R cDNAs (1–545 and 1–537) were prepared using PCR. The PCR products were cloned into the pCR2.1

vector (Invitrogen), and the nucleotide sequences were verified using BigDye Terminator v3.1 Cycle Sequencing Kit and ABI PRISM 3100 Genetic Analyzer (both from Applied Biosystems). The insert cDNAs were subcloned into the pcDNA3.1 expression vector (Invitrogen). The plasmid of inositol polyphosphate 5-phosphatase type IV was prepared as described previously (46). The GFP-PLCδ-PH plasmid was obtained through Addgene (#21179).

### Immunofluorescence and live-cell imaging

Immunofluorescence was performed as described previously (33). In brief, cells were fixed in 4% paraformaldehyde, permeabilized with 0.1% Triton X-100, and stained with anti-mouse CSF1R (AFS98; BioLegend) or anti-human CSF1R antibodies (9-4D2-1E; BioLegend) followed by Alexa Fluor 488–labeled anti-rat IgG or Alexa Fluor 633–labeled anti-rat IgG (both from Molecular Probes). Anti-TNFAIP2 antibody (F-6; Santa Cruz Biotechnology) or anti-CSF-2 receptor α chain antibody (698423; R&D Systems) was also used. DAPI (Molecular Probes), Alexa Fluor 633–conjugated WGA (Molecular Probes), and anti-GM130 antibody (D6B1; Cell Signaling Technology) were also used to visualize nuclei, the plasma membrane, and the Golgi, respectively. Signals were visualized using a confocal laser-scanning microscope FV1200 (Olympus) or TCS SP8 (Leica). Image processing was performed using FV viewer ver. 4.2 software (Olympus) or LAS X software ver.1.4.5 (Leica). The signal of CSF1R aggregates was quantified using ImageJ 1.52n software (NIH). To visualize PIP2 in living cells, the signal of GFP-PLCδ-PH in unfixed 293 cells was analyzed using TCS SP8.

### Flow cytometry

The cell surface expression of CSF1R was analyzed by flow cytometry (59). In brief, cells were detached using enzyme-free cell dissociation buffer (Millipore), stained with fluorescent dye–labeled antibodies, and analyzed on FACSCanto II (BD Biosciences) or CytoFLEX (Beckman Coulter) using FlowJo software (FlowJo LLC). In selected experiments, the intracellular level of CSF1R was also analyzed by flow cytometry. In brief, cells were detached, fixed (Fixation Buffer; BioLegend), permeabilized (Intracellular Staining Permeabilization Wash Buffer; BioLegend), and stained with fluorescent dye–labeled antibodies. The antibodies used were as follows: allophycocyanin (APC)-labeled anti-mouse CSF1R (AFS98) and APC-labeled anti-human CSF1R (9-4D2-1E4) (both from BioLegend).

### Western blotting

Western blotting was performed as described previously (32). In brief, cells were lysed with Nonidet P-40 lysis buffer containing protease and phosphatase inhibitors. In selected experiments, the total cell lysates were incubated with anti-phosphotyrosine antibody (PY99; Santa Cruz Biotechnology) or anti-Flag antibody (M2; Sigma-Aldrich) (to precipitate Flag-tagged TNFAIP2) followed by Protein A/G PLUS Agarose (Santa Cruz Biotechnology) (20). The total cell lysates, the anti-phosphotyrosine immunoprecipitates, or the anti-Flag immunoprecipitates were subjected to Western blotting. The antibodies used were as follows: anti-phospho-ERK (E-4) and

anti-total ERK (C-9) (both from Santa Cruz Biotechnology), anti-phospho-p38 (3D7; Cell Signaling Technology), anti-total p38 (A-12; Santa Cruz Biotechnology), anti-phospho-JNK (81E11) and anti-total JNK (#4672) (both from Cell Signaling Technology), anti-phosphotyrosine (PY99; Santa Cruz Biotechnology), anti-CSF1R (#3152; Cell Signaling Technology) (to detect mouse CSF1R), anti-CSF1R (E7S2S; Cell Signaling Technology) (to detect human CSF1R), anti-phospho-CSF1R (Tyr809; Cell Signaling Technology), anti-TNFAIP2 (F-6; Santa Cruz Biotechnology), anti-phospho-Stat5 (Stat5A Tyr694/Stat5B Tyr699; Proteintech), anti-Stat5 (#610191; BD Biosciences), and anti-actin (EPR16769; Abcam). Detection was performed using HRP-labeled secondary antibodies (GE Healthcare), Western blot ultrasensitive HRP substrate (TaKaRa-Bio), and ImageQuant LAS 4000 image analyzer (GE Healthcare). The density of the upper or lower band of CSF1R in RAW264.7 cells was quantified using ImageJ software after normalization to the density of the actin band.

### Nitric oxide production and iNOS mRNA expression

The concentration of nitric oxide in the culture supernatants of RAW264.7 cells was quantified using the Griess reagent (Dojindo). The expression of iNOS mRNA was analyzed by real-time RT–PCR (59). In brief, RNA was isolated using ISOGEN II reagent (Nippon Gene), cDNA was prepared using M-MLV RT (Invitrogen), and real-time PCR was performed using SYBR Premix Ex Taq II (TaKaRa-Bio) and LightCycler (Roche). β-Actin mRNA was quantified as an internal control. The primer pairs were as follows: 5′-ACCTTGTTCAGC-TACGCCTT-3′ and 5′-CATTCCCAAATGTGCTTGTC-3′ (iNOS), and 5′-CATCCGTAAAGACCTCTATGCCAAC-3′ and 5′-ATGGAGCCACCGATCCACA-3′ (mouse β-actin).

### TNFAIP2 and CSF1R mRNA expression

The expression of mouse TNFAIP2 mRNA, human TNFAIP2 mRNA, or human CSF1R mRNA was also analyzed by real-time RT–PCR (59), as above. The primer pairs were as follows: 5′-GTCAACATCATGGCCAA-CATCA-3′ and 5′-CTCTGGAGCAGGGTGTCGAA-3′ (mouse TNFAIP2), 5′-CGACACCTACATGCTG-3′ and 5′-CGAGCCCCATACCCTG-3′ (human TNFAIP2), 5′-GGCGTCGACTATAAGAACATCCA-3′ and 5′-GAGACAGGCCTCATCTCCACA-3′ (human CSF1R), and 5′-TGACGGGGTCACCCACACTG-3′ and 5′-AAGCTG-TAGCCGCGCTCGGT-3′ (human β-actin).

### Phagocytosis assays

Phagocytic activity was quantified as described previously (59). In brief, RAW264.7 cells were incubated with fluorescent microspheres (Fluoresbrite carboxylate microspheres with a 0.7 μm diameter, Polysciences) for 2 h and analyzed by flow cytometry as described above.

### MTT assay

The number of cells was assessed using MTT reagent (32). The absorbance of the wells was measured at 595 nm.

## Statistical analysis

Differences between the two groups were analyzed by an unpaired *t* test. Differences between multiple groups were analyzed by a one-way or two-way ANOVA. All statistical analyses were conducted using PRISM 10 (GraphPad).

# Supplementary Information

# Acknowledgements

We thank K Nasu and I Suzu for their technical and secretarial assistance, respectively. This study was supported by grants (KAKENHI) from the Japan Society for the Promotion of Science (JSPS) (18K19457 to S Suzu, 23H02727 to S Suzu, and 18K07155 to M Hiyoshi), a grant from the Japan Agency for Medical Research and Development (AMED) (19fk0410018h0002 to M Hiyoshi), and a grant from the Japan Science and Technology Agency (JST) SPRING (JPMJSP2127 to RA Abdelnaser).

## Author Contributions

RA Abdelnaser: resources, funding acquisition, investigation, and writing—original draft.
M Hiyoshi: resources, funding acquisition, and investigation.
N Takahashi: investigation.
YM Eltalkhawy: investigation.
H Mizuno: methodology.
S Kimura: resources and writing—review and editing.
K Hase: resources and writing—review and editing.
H Ohno: resources and writing—original draft.
K Monde: resources and writing—review and editing.
A Ono: resources and writing—review and editing.
S Suzu: conceptualization, funding acquisition, and writing—original draft.

## Conflict of Interest Statement

The authors declare that they have no conflict of interest.

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
