## [Reviewer comments · Life Science Alliance]

Life Science Alliance

Identification of TNFAIP2 as a unique cellular regulator of CSF-1 receptor activation

Randa Abdelnaser, Masateru Hiyoshi, Naofumi Takahashi, Youssef Eltalkhawy, Hidenobu Mizuno, Shunsuke Kimura, Koji Hase, Hiroshi Ohno, Kazuaki Monde, Akira Ono, and Shinya Suzu

DOI: <https://doi.org/10.26508/lsa.202403032>

Corresponding author(s): Shinya Suzu, Kumamoto University

Review Timeline:

Submission Date:	2024-09-05
Editorial Decision:	2024-10-21
Revision Received:	2025-01-11
Editorial Decision:	2025-02-04
Revision Received:	2025-02-04
Accepted:	2025-02-05

Transaction Report:

October 21, 2024

Re: Life Science Alliance manuscript #LSA-2024-03032-T

Prof. Shinya Suzu
Kumamoto University
Center for AIDS Research
2-2-1 Honjo
Kumamoto, Kumamoto 860-0811
Japan

Dear Dr. Suzu,

Thank you for submitting your manuscript entitled "Identification of M-Sec as a unique cellular regulator of CSF-1 receptor activation". The manuscript has been evaluated by expert reviewers, whose reports are appended below. Unfortunately, after an assessment of the reviewer feedback, our editorial decision is against publication in Life Science Alliance.

Although your manuscript is intriguing, I feel that the points raised by the reviewers are more substantial than can be addressed in a typical revision period. If you wish to expedite publication of the current data, it may be best to pursue publication at another journal.

Given the interest in the topic, I would be open to re-submission to Life Science Alliance of a significantly revised and extended manuscript that fully addresses the reviewers' concerns and is subject to further peer review. If you would like to resubmit this work to Life Science Alliance, you may submit an appeal directly through our manuscript submission system. Please note that priority and novelty would be reassessed at re-submission.

Regardless of how you choose to proceed, we hope that the comments below will prove constructive as your work progresses.

Thank you for thinking of Life Science Alliance as an appropriate place to publish your work.

Sincerely,

Reviewer #1 (Comments to the Authors (Required)):

This paper reports that the protein M-Sec mediates Fms aggregation and the absence of this interaction facilitates Fms activation and signaling. The interaction is reported to be mediated by PIP2. The paper contains many figures showing essentially similar findings in different models of CSF1R/TNFAIP2 over-expression/inhibition/knockdown.

Comments and questions:

- Please use official gene symbols: CSF1R and TNFAIP2
- The papers cited in support of the fact that CSF1R monomers form large aggregates do not actually support this. Ref 21 speculates this may be the case. Ref 23 concerns nuclear CSF1R.
- Are the aggregates in a specific cellular compartment, such as the golgi, where CSF1R has previously been localized? - and TNFAIP is also localised? <https://www.sciencedirect.com/science/article/pii/S0898656816301140>
- Figure 1 - there is apparently no surface CSF1R expression? - how was the cell surface defined for quantitation? are these cells CSF1 starved - which would bring the receptor to the surface. What is specificity of the M-Sec inhibitor and the efficiency of knock-down?
- Figure 2 - CSF1-dependent induction of iNOS is unusual, typically LPS/IFN γ stimulation are required - please comment. The text mentions that M-Sec knockdown did not affect LPS stimulated iNOS expression but data are not shown. This should be shown, as LPS strongly induces M-Sec/Tnfaip2.
- Figure 3 - p38 and JNK are not classical pathways downstream of CSF2 - please comment
- Figure 6 - no control staining is shown (ie 293 without Fms expression)
- Figure 10 - how is this live cell imaging? what happened in M-Sec/Fms co-expressing cells

Reviewer #2 (Comments to the Authors (Required)):

Fms aggregation results from low affinity interactions between unliganded CSF-1Rs that contributes to the ligand-induced dimerization. This study nicely shows the importance of M-Sec for Fms aggregate formation, which, in itself, is an important finding. However, at this stage, in contrast to the authors' claim, the mechanism is not clear. The authors suggest that M-Sec brings the Fms monomers close to each other via their binding to PIP2. However, while this is possible, they have not provided direct evidence for it, nor eliminated the possibility of alternative mechanisms, such as direct association of M-Sec with Fms. In particular, it is not clear that the comparison between the I-537 and I-545 mutants is valid unless the I-537 mutant is shown to be expressed at the cell surface.

P2 Line 6: Change "of reduced " to "of M-Sec reduced"

P2 line 10. Change "CSF-1" to "CSF-1R"

Results:

Fig. 1 legend. It is important to know the concentrations of CSF-1 the cells were incubated in, prior to the 48 h incubation with and without M-Sec. Also, indicate whether CSF-1 was present or absent during the 48 h incubation with DMSO or M-Sec and at what concentration. If absent, indicate in the Materials and Methods how long the bone marrow-derived macrophages survive in the absence of CSF-1.

Figures 3 & 4 can be combined.

P6 Line 13. Replace "cancelled" with "rescued"

P7 Lines 5- 9. The interpretation here is not entirely correct. According to ref 33 and earlier work from that group, the initial apparent decrease in the mature form of the CSF-1R is due to its rapid polyubiquitination, causing an increased spreading of tyrosine phosphorylated receptors that therefore don't stain as well with anti-CSF-1R antibody. It is subsequently degraded intralysosomally.

P7 Line 18. Replace "in 293 cells" with "in Fms-transfected 293 cells"

Fig. 6 Legend. A-D. Replace "The parent or M-Sec-expressing 293 cells" with "The parent or M-Sec-expressing Fms-transfected 293 cells"

P8. Lines 10-17. There is no conclusion presented from this experiment. In addition, the I-537 mutant lacks the "stop transfer" sequence that prevents the receptor entering the cytoplasm during synthesis. Therefore, the I-537 mutant is likely only expressed in the cytoplasm and not at the cell surface. In order to compare aggregation phenomena between the two they should both be expressed at the cell surface. Their cell surface expression is not shown and, from the image in Figure 1B, it seems unlikely that I-537 exhibits cell surface expression. If so, the comparison is not valid. It is possible that M-Sec directly associates with Fms. Their possible direct association could be examined when both are detergent solubilized.

P32. Fig.8 Legend title. Change "-expressing 293 cells" to "-expressing Fms-transfected 293 cells"

P32. Fig. 8B legend. Specify Fms-transfected 293 cells were used.

Figs. 6-9. Legends. Where it is stated that cells were cultured for 2 days, specify in the presence or absence of CSF-1.

Fig. 10 legend and text. Specify whether the parent or M-Sec-expressing 293 cells are Fms-transfected or not and indicate that duplicate panels shown.

Reviewer #3 (Comments to the Authors (Required)):

M-Sec is a 74 kDa protein expressed in myeloid cells of both mouse and human origin, including dendritic cells and macrophages, and in a subtype of enterocytes (M cells) that has homology to a component of the exocyst complex, Sec6. The corresponding gene was identified in 1994 as a tumor necrosis factor-alpha-inducible primary response gene (Tnfaip2) and designated also as B94. This protein was involved in the formation of membrane protrusions extending from the plasma membrane, including tunneling nanotubes. The manuscript by Randa A Abdelnaser et al identifies M-Sec as a partner of Colony Stimulating Factor 1 receptor (Csf1r, also known as Fms) that favors its activation through the formation of aggregates in macrophages, the two proteins interacting with PIP2 (phosphatidylinositol 4,5-biphosphate), which is supposed to drive the formation of aggregates. This limited study, almost exclusively performed in cell lines, is nevertheless clearly reported and the proposed conclusions summarize the results obtained in cell lines. The description of M-Sec /PIP2 pathway to Csf1r signaling, would have been more significant by analyzing the phenotype of M-Sec KO mice. The discussion should better address the

potential significance of the described regulation.

Main comments

- A key question that cannot be addressed in this study is the functional significance of Csf1r aggregates. As described, Csf1-induced signaling is inhibited by deletion or inhibition of M-Sec. Homozygous M-Sec-KO mice are viable and grow normally, nevertheless develop renal functional alterations worsening with age (PMID: 33722931). Inhibition of Csf1r pathway, for example with therapeutic antibodies, leads to relatively rapid depletion of most tissue macrophage populations and osteoclasts. What is the functional importance of the M-Sec involving pathway? For example, do monocytes that survive and differentiate in liquid culture upon Csf1 exposure die upon inhibition of M-Sec?
- References 21 to 23 are mentioned but these manuscripts describe Csf1r interactions with other cellular components in response to the ligand or in the nucleus but do not mention such aggregates. Can we observe such aggregates in monocytes as well? in cancer cells expressing Csf1r? Are these aggregates an absolute requirement for Csf1 signaling?

Minor

- The introduction should better prepare the reader by describing M-Sec discovery, functions, manipulation (including the phenotype of ko mice) and inhibition in more details
- The name and concentration of the tested M-Sec inhibitor as well as duration of cell treatment should be indicated in both the results section and the legend of Figure 1. Do the cell die at later time point upon exposure? Did the author test a dose-dependent effect?
- Figure 2D, M-Sec knock-down appears to significantly sensitize macrophages to CSF-2-driven phagocytosis. The authors do not mention nor comment this observation that correlates with an increased signaling on Figure 3 (p-ERK and p-p38 at 1 min).
- It would be useful to indicate to which extent the formation of Fms aggregates is a characteristic of any cell type expressing Csf1r/Fms.

Reviewer #1

This paper reports that the protein M-Sec mediates Fms aggregation and the absence of this interaction facilitates Fms activation and signaling. The interaction is reported to be mediated by PIP2. The paper contains many figures showing essentially similar findings in different models of CSF1R/TNFAIP2 over-expression/inhibition/knockdown.

Please notice the following changes in the new version:

- (1) We used "CSF1R" in place of "Fms" and "TNFAIP2" in place of "M-Sec" in the new version according to your comment.
- (2) The previous Figs 5, 6, 7, 8, 9 and 10 are Figs 4, 5, 6, 7, 8 and 9 in the new version because Figs 3 and 4 were combined according to the comment of the reviewer #2.
- (3) We used "CSF-1-expanded self-renewing bone marrow-derived macrophages" (CSF-1-expanded self-renewing BM-M Φ ; Reference #34) in place of "bone-marrow-derived macrophages" to more clearly specify the cells that we employed in this study.

Comment 1. Please use official gene symbols: CSF1R and TNFAIP2

Reply

According to the comment, we used CSF1R and TNFAIP2 throughout the manuscript.

Comment 2. The papers cited in support of the fact that CSF1R monomers form large aggregates do not actually support this. Ref 21 speculates this may be the case. Ref 23 concerns nuclear CSF1R.

Reply
Modified from Figure 1 of Reference #22

The biochemical analysis of Reference #21 raised the possibility that CSF1R monomers form aggregates in the mouse macrophage cell line BAC1.2F5. The formation of CSF1R

aggregates was not the main topic of References #22 and #23, but the immunofluorescence of these studies showed that CSF1R was detected as aggregate-like signals in BAC1.2F5 cells, as shown above (the aggregate-like CSF1R signals are indicated by yellow arrows). To more accurately explain these points, we rewrote the sentences as follows (underlined):

[Page 4, lines 6 - 9 (Introduction)]

Interestingly, the biochemical analysis raised the possibility that CSF1R monomers are clustered and form aggregates in the mouse macrophage cell line BAC1.2F5 (21). Consistent with this, CSF1R was detected as aggregate-like signals in BAC1.2F5 cells in the immunofluorescence analysis (22, 23).

Comment 3. Are the aggregates in a specific cellular compartment, such as the golgi, where CSF1R has previously been localized? - and TNFAIP is also localised? <https://www.sciencedirect.com/science/article/pii/S0898656816301140>

Reply

According to the comments, we performed additional experiments using the CSF-1-expanded self-renewing BM-MΦ. As shown in Fig S2, several CSF1R aggregates, but not all, localized to the Golgi. As shown in Fig S3, TNFAIP2 diffusely localized throughout the cytoplasm (the similar localization is observed in TNFAIP2-expressing 293 cells, as shown in Fig S12). The experimental procedures were described in the legends of Figs S2 and S3, and the Materials and Methods section (page 21, lines 3 - 8). To explain the results in Figs S2 and S3, we added the following sentences (underlined):

[Page 6, lines 11 - 12 (Results)]

Several CSF1R aggregates, but not all, localized to the Golgi (Fig S2). TNFAIP2 diffusely localized throughout the cytoplasm (Fig S3),

Comment 4-1. Figure 1 - there is apparently no surface CSF1R expression? - how was the cell surface defined for quantitation? are these cells CSF1 starved - which would bring the receptor to the surface. What is specificity of the M-Sec inhibitor and the efficiency of knock-down?

Reply

According to the comment, we performed an additional immunofluorescence using the CSF-1-expanded self-renewing BM-MΦ, in which we used wheat germ agglutinin (WGA) to visualize the plasma membrane and overlaid the images composed of serial Z-actions. As shown in Fig S1, consistent with the flow cytometric analysis (Fig 1C), the CSF1R aggregates were detectable at the plasma membrane. The experimental

procedures were described in the legend of Fig S1, and the Materials and Methods section (page 21, lines 5 - 8). To explain the result in Fig S1, we added the following sentence (underlined):

[Page 6, lines 10 and 11 (Results)]

The CSF1R aggregates were detectable at the plasma membrane (Fig S1).

Comment 4-2. Figure 1 - there is apparently no surface CSF1R expression? - how was the cell surface defined for quantitation? are these cells CSF1 starved - which would bring the receptor to the surface. What is specificity of the M-Sec inhibitor and the efficiency of knock-down?

Reply

Thank you for pointing out our careless mistakes.

(1) The cell surface expression level of CSF1R in Figs 1C and 1F was quantified by flow cytometry. We corrected the legends of Fig 1C and 1F.

(2) When the CSF-1-expanded self-renewing BM-M Φ were used for CSF1R flow cytometry and CSF1R immunofluorescence, they were CSF-1 starved for 6 hours before those analyses. We specified the point in the legends of Fig 1 and other related figures, including Figs S1 and S2.

Comment 4-3. Figure 1 - there is apparently no surface CSF1R expression? - how was the cell surface defined for quantitation? are these cells CSF1 starved - which would bring the receptor to the surface. What is specificity of the M-Sec inhibitor and the efficiency of knock-down?

Reply

According to the comment, we examined whether the inhibitor affects the distribution of cell surface receptors other than CSF1R, using the CSF-1-expanded self-renewing BM-M Φ . To exclude the possibility that the observed inhibitory effect of the inhibitor on the CSF-1 - CSF1R axis is due to non-specific cytotoxicity, we choose CSF-2 receptor α chain because the inhibitor did not affect the response of macrophages to CSF-2 (iNOS mRNA upregulation, please see Fig 2A). As shown in Fig S4, there was no obvious effect of the TNFAIP2 inhibitor on the cellular distribution of CSF-2 receptor α chain, although the signal was observed at nuclei in addition to the cell surface. The experimental procedures were described in the legend of Fig S4, and the Materials and Methods section (page 21, line 4). To explain the result in Fig S4, we added the following sentence (underlined):

[Page 6, lines 17 - 19 (Results)]

In contrast, there was no obvious effect of the TNFAIP2 inhibitor on the cellular distribution of CSF-2 receptor α chain in the CSF-1-expanded self-renewing BM-M Φ (Fig S4).

Comment 4-4. Figure 1 - there is apparently no surface CSF1R expression? - how was the cell surface defined for quantitation? are these cells CSF1 starved - which would bring the receptor to the surface. What is specificity of the M-Sec inhibitor and the efficiency of knock-down?

Reply

Modified from Figure 1 of Reference #26

The TNFAIP2 knockdown efficiency in RAW264.7 cells was previously assessed by western blotting, as shown above. We further confirmed the knockdown efficiency by qRT-PCR, as shown in Fig S5. The experimental procedures were described in the legend of Fig S5, and the Materials and Methods section (page 23, lines 7 - 15). To explain the result in Fig S5, we added the following words (underlined):

[Page 6, lines 20 and 21]

We next compared the control and TNFAIP2 knockdown mouse macrophage cell line RAW264.7 that were established previously (26) (Fig S5).

Comment 5-1. Figure 2 - CSF1-dependent induction of iNOS is unusual, typically LPS/IFN \$\gamma\$ stimulation are required - please comment. The text mentions that M-Sec knockdown did not affect LPS stimulated iNOS expression but data are not shown. This should be shown, as LPS strongly induces M-Sec/Tnfaip2.

Reply

- (1) In RAW264.7 cells, CSF-1 can induce iNOS expression, as shown in this study and another group's study (Lin CW, et al, Carcinogenesis 31: 2039-2048, 2010).
- (2) In this study, we used the CSF-1-expanded self-renewing mouse bone marrow-derived macrophages (CSF-1-expanded self-renewing BM-M Φ), which were established through

the long-term culture of bone marrow cells in the presence of CSF-1 with repeated passages (Reference #34). As shown in Fig S7, CSF-1 upregulated iNOS mRNA expression in the cells when combined with 1 or 10 ng/ml LPS. [The experimental procedures were described in the legend of Fig S7, and the Materials and Methods section (page 19, lines 7 - 9).]

However, as pointed out, unlike typical macrophages, CSF-1 alone was enough to upregulate iNOS mRNA expression at a detectable level in the cells (Fig S7 and Fig 2). This is presumably because of their adaptation to CSF-1 during the long-term culture with CSF-1. In fact, CSF-1, but not CSF-2, efficiently supports their stable proliferation (Reference #34). To explain these points and the result in Fig S7, we added the following sentences (underlined):

[Page 6, lines 4 - 8 (Results)]

We initially analyzed the CSF-1-expanded self-renewing mouse bone marrow-derived macrophages (hereinafter referred to as CSF-1-expanded self-renewing BM-MΦ) (34). These macrophages were established through the long-term culture of bone marrow cells in the presence of CSF-1 with repeated passages, and stably proliferated in the presence of CSF-1 (34).

[Page 7, lines 2 - 4 (Results)]

The CSF-1-expanded self-renewing BM-MΦ proliferated in the presence of CSF-1, but not of CSF-2 (34), presumably because of their adaptation to CSF-1 during the long-term culture with CSF-1.

[Page 7, lines 8 - 15 (Results)]

It is well known that CSF-2 or lipopolysaccharide (LPS) upregulates the expression of inducible nitric oxide synthase (iNOS) and that CSF-1 upregulates the iNOS expression when combined with LPS in typical macrophages. In fact, CSF-1 upregulated iNOS mRNA expression in the CSF-1-expanded self-renewing BM-MΦ when combined with 1 or 10 ng/ml LPS (Fig S7). However, unlike typical macrophages, CSF-1 alone was enough to upregulate iNOS mRNA expression at a detectable level in the CSF-1-expanded self-renewing BM-MΦ (Fig S7), presumably because of their adaptation to CSF-1 during the long-term culture with CSF-1.

Comment 5-2. Figure 2 - CSF1-dependent induction of iNOS is unusual, typically LPS/IFNγ stimulation are required - please comment. The text mentions that M-Sec knockdown did not affect LPS stimulated iNOS expression but data are not shown. This should be shown, as LPS strongly induces M-Sec/Tnfaip2.

Reply

According to the comment, we showed the results in Figs 2B (iNOS) and 2C (nitric oxide). The experimental procedures are described in the legend. In the experiments, we used the suboptimal concentration of LPS (1 ng/ml), which upregulated iNOS mRNA expression and nitric oxide production at levels similar to that by CSF-1 or CSF-2.

Comment 6. Figure 3 - p38 and JNK are not classical pathways downstream of CSF2 - please comment

Reply

p38 and JNK can be activated by CSF-2, but they are not classical pathways downstream of CSF-2, as pointed out. Thus, we analyzed Stat5, the well-known downstream molecule of CSF-2 signaling. As shown in Fig 3B, the TNFAIP2 knockdown did not affect the CSF-2-mediated activation of Stat5. The experimental procedures were described in the legend of Fig 3B, and the Materials and Methods section (page 22, lines 14 and 15). To explain the new result in Fig 3B, we added the following sentence (underlined):

[Page 8, lines 9 and 10 (Results)]

In fact, CSF-2 activated Stat5 in both the control and TNFAIP2 knockdown cells at the similar level (Fig 3B, p-Stat5 blot).

Comment 7. Figure 6 - no control staining is shown (ie 293 without Fms expression)

Reply

[The previous Fig 6 is Fig 5 in this version]

According to the comment, we showed the control images (Fig 5A, left panels). The experimental procedures were described in the legend of Fig 5A.

Comment 8. Figure 10 - how is this live cell imaging? what happened in M-Sec/Fms co-expressing cells

Reply

[The previous Fig 10 is Fig 9 in this version]

According to the comment, we performed an additional experiment. As shown in Fig S9, the large GFP-PLC δ -PH-positive structures were also observed in 293 cells co-expressing TNFAIP2 and CSF1R, but not in 293 cells expressing CSF1R alone. The experimental procedures were described in the legend, and the Materials and Methods section (page 19 line 25 - page 20, line 2). To explain the result in Fig S9, we added the following sentence (underlined):

[Page 12, lines 6 - 8 (Results)]

Such large GFP-PLC δ -PH-positive structures were also observed in 293 cells co-expressing TNFAIP2 and CSF1R, but not in 293 cells expressing CSF1R alone (Fig S9).

Reviewer #2

Fms aggregation results from low affinity interactions between unliganded CSF-1Rs that contributes to the ligand-induced dimerization. This study nicely shows the importance of M-Sec for Fms aggregate formation, which, in itself, is an important finding. However, at this stage, in contrast to the authors' claim, the mechanism is not clear. The authors suggest that M-Sec brings the Fms monomers close to each other via their binding to PIP2. However, while this is possible, they have not provided direct evidence for it, nor eliminated the possibility of alternative mechanisms, such as direct association of M-Sec with Fms. In particular, it is not clear that the comparison between the I-537 and I-545 mutants is valid unless the I-537 mutant is shown to be expressed at the cell surface.

Please notice the following changes in the new version:

- (1) We used "CSF1R" in place of "Fms" and "TNFAIP2" in place of "M-Sec" in the new version according to the comment of the reviewer #1.
- (2) The previous Figs 5, 6, 7, 8, 9 and 10 are Figs 4, 5, 6, 7, 8 and 9 in the new version because Figs 3 and 4 were combined according to your comment.
- (3) We used "CSF-1-expanded self-renewing bone marrow-derived macrophages" (CSF-1-expanded self-renewing BM-MΦ; Reference #34) in place of "bone-marrow-derived macrophages" to more clearly specify the cells that we employed in this study.

Comment 1. P2 line 6. Change "of reduced " to "of M-Sec reduced"

Reply

Thank you for pointing out our careless mistake. We corrected the sentence.

Comment 2. P2 line 10. Change "CSF-1" to "CSF-1R"

Reply

Thank you for pointing out our careless mistake. We corrected the sentence.

Comment 3. Fig. 1 legend. It is important to know the concentrations of CSF-1 the cells were incubated in, prior to the 48 h incubation with and without M-Sec. Also, indicate whether CSF-1 was present or absent during the 48 h incubation with DMSO or M-Sec and at what concentration. If absent, indicate in the Materials and Methods how long the bone marrow-derived macrophages survive in the absence of CSF-1.

Reply

The CSF-1-expanded self-renewing BM-MΦ were maintained with 100 ng/ml CSF-1, and treated with DMSO or the TNFAIP2 inhibitor in the presence of 100 ng/ml CSF-1

for 48 hours. Then, these cells were CSF-1 starved for 6 hours and analyzed by the CSF1R immunofluorescence and CSF1R flow cytometry. We specified these points in the legend of Fig 1A/B/C.

Comment 4. Figures 3 & 4 can be combined.

Reply

According to the suggestion, we combined Figs 3 and 4.

Comment 5. P6 line 13. Replace "cancelled" with "rescued"

Reply

According to the comment, we corrected the sentence (page 8, line 14, in this version).

Comment 6. P7 lines 5-9. The interpretation here is not entirely correct. According to ref 33 and earlier work from that group, the initial apparent decrease in the mature form of the CSF-1R is due to its rapid polyubiquitination, causing an increased spreading of tyrosine phosphorylated receptors that therefore don't stain as well with anti-CSF-1R antibody. It is subsequently degraded intralysosomally.

Reply

According to the thoughtful comment, we rewrote the sentences as follows (underlined):
[Page 9, lines 8 - 11 (Results)]

This is because the upper band is the fully *N*-glycosylated cell surface form whereas the lower is the hypo-*N*-glycosylated immature intracellular form (37), and the activation of the mature CSF1R associates with its rapid polyubiquitination and apparent decrease in western blotting (38).

Comment 7. P7 Line 18. Replace "in 293 cells" with "in Fms-transfected 293 cells"

Reply

According to the comment, we corrected the sentence (page 9, lines 21 and 22, in this version).

Comment 8. Fig. 6 Legend. A-D. Replace "The parent or M-Sec-expressing 293 cells" with "The parent or M-Sec-expressing Fms-transfected 293 cells"

Reply

According to the comment, we rewrote the legend of the figure (the previous Fig 6 is Fig 5 in this version).

Comment 9-1. P8 lines 10-17. There is no conclusion presented from this experiment. In addition, the I-537 mutant lacks the "stop transfer" sequence that prevents the receptor entering the cytoplasm during synthesis. Therefore, the I-537 mutant is likely only expressed in the cytoplasm and not at the cell surface. In order to compare aggregation phenomena between the two they should both be expressed at the cell surface. Their cell surface expression is not shown and, from the image in Figure 1B, it seems unlikely that I-537 exhibits cell surface expression. If so, the comparison is not valid. It is possible that M-Sec directly associates with Fms. Their possible direct association could be examined when both are detergent solubilized.

Reply

We appreciate the comment. As shown in Fig S8, the 1-537 mutant was expressed on the surface of the parent 293 cells, the level of which was higher than that of the wild-type CSF1R or comparable to that of the 1-545 mutant (Fig S8A, upper panels). Thus, the inability of the 1-537 mutant to form aggregates in the presence of TNFAIP2 might be unrelated to its intrinsic cell surface expression level. Interestingly, the TNFAIP2 expression increased the cell surface expression level of the wild-type and the 1-545 mutant (Fig S8A, lower panels), which was consistent with their decreased intracellular level (Fig S8B). Such changes were not observed for the 1-537 mutant (Fig S8). Although further studies are necessary to understand the underlying mechanism, these results suggest that the formation of CSF1R aggregates in the presence of TNFAIP2 is associated with the intracellular-to-cell surface distribution of CSF1R, and raise the possibility that the putative PIP2-binding motif in the juxtamembrane region CSF1R is involved in the processes. The experimental procedures of Fig S8 were described in the legend, and the Materials and Methods section (page 21, lines 18 - 22). To explain the results in Fig S8, we added the following sentences (underlined):

[Page 10, line 21 - page 11, line 3 (Results)]

The 1-537 mutant was expressed on the surface of the parent 293 cells, the level of which was higher than that of the wild-type CSF1R or comparable to that of the 1-545 mutant (Fig S8A, upper panels). Interestingly, the TNFAIP2 expression increased the cell surface expression level of the wild-type and the 1-545 mutant (Fig S8A, lower panels), which was consistent with their decreased intracellular level (Fig S8B). Such changes were not observed for the 1-537 mutant (Fig S8). These results suggest that the formation of CSF1R aggregates in the presence of TNFAIP2 is associated with the intracellular-to-cell surface distribution of CSF1R, and raise the possibility that the putative PIP2-binding motif in the juxtamembrane region CSF1R is involved in the processes.

Comment 9-2. P8 lines 10-17. There is no conclusion presented from this experiment. In addition, the I-537 mutant lacks the "stop transfer" sequence that prevents the receptor entering the cytoplasm during synthesis. Therefore, the I-537 mutant is likely only expressed in the cytoplasm and not at the cell surface. In order to compare aggregation phenomena between the two they should both be expressed at the cell surface. Their cell surface expression is not shown and, from the image in Figure 1B, it seems unlikely that I-537 exhibits cell surface expression. If so, the comparison is not valid. It is possible that M-Sec directly associates with Fms. Their possible direct association could be examined when both are detergent solubilized.

Reply

According to the comment, we performed the co-immunoprecipitation using TNFAIP2-expressing CSF1R-transfected 293 cells. As shown in Fig S11, the wild-type CSF1R and the 1-545 mutant, but not the 1-537 mutant, were detected in the anti-TNFAIP2 immunoprecipitates, which was consistent with the finding that the wild-type CSF1R and the 1-545 mutant, but not the 1-537 mutant, formed aggregates in the presence of TNFAIP2. However, it should be mentioned that TNFAIP2 does not necessarily co-localize with CSF1R since it diffusely localizes throughout the cytoplasm in the transfected 293 cells (Fig S12). The similar diffuse localization was also observed in the CSF-1-expanded self-renewing BM-MΦ (Fig S3). Therefore, we feel that it remains unclear to what degree the possible TNFAIP2 - CSF1R interaction contributes to the formation of CSF1R aggregates by TNFAIP2. We also feel that it is necessary to clarify whether TNFAIP2 directly or indirectly interacts with CSF1R. The experimental procedures of Figs S11 and S12 were described in the legends, and the Materials and Methods section (page 22, lines 1- 5). To explain the results in Figs S11 and S12, we added the following sentences (underlined):

[Page 16, lines 8 - 18 (Discussion)]

It is possible that the formation of CSF1R aggregates by TNFAIP2 is due to their interaction. In our co-immunoprecipitation using TNFAIP2-expressing CSF1R-transfected 293 cells, the wild-type CSF1R and the 1-545 mutant, but not the 1-537 mutant, were detected in the anti-TNFAIP2 immunoprecipitates (Fig S11), which was consistent with the finding that the wild-type CSF1R and the 1-545 mutant, but not the 1-537 mutant, formed aggregates in the presence of TNFAIP2. However, it should be mentioned that TNFAIP2 does not necessarily co-localize with CSF1R since it diffusely localizes throughout the cytoplasm in the transfected 293 cells (Fig S12). Therefore, it remains unclear to what degree the possible TNFAIP2 - CSF1R interaction contributes to the formation of CSF1R aggregates by TNFAIP2. It is also necessary to clarify whether

TNFAIP2 directly or indirectly interacts with CSF1R.

Comment 10. P32. Fig. 8 legend title. Change "-expressing 293 cells" to "-expressing Fms-transfected 293 cells"

Reply

According to the comment, we corrected the title (the previous Fig 8 is Fig 7 in this version).

Comment 11. P32. Fig. 8B legend. Specify Fms-transfected 293 cells were used.

Reply

In the panels C and D, we used CSF1R-transfected 293 cells, as specified in the legend. In the panel B, we used un-transfected parent 293 cells and un-transfected TNFAIP2-expressing 293 cells because the purpose of this experiment was to confirm that the expression level of the TNFAIP2 mutants ($\Delta K1$ and $\Delta K2$) was comparable to that of the wild-type. We specified the point in the legend (the previous Fig 8 is Fig 7 in this version).

Comment 12. Figs. 6-9. Legends. Where it is stated that cells were cultured for 2 days, specify in the presence or absence of CSF-1.

Reply

The cells were transfected with CSF1R plasmid and cultured for 2 days in the absence of CSF-1. We specified this point in the legends of these figures (the previous Figs 6-9 is Figs 5-8 in this version).

Comment 13. Fig. 10 legend and text. Specify whether the parent or M-Sec-expressing 293 cells are Fms-transfected or not and indicate that duplicate panels shown.

In the previous Fig 10 (Fig 9 in this version), we used the parent and TNFAIP2-expressing 293 cells, and showed the duplicate panels. We specified these points in the legend of Fig 9 and the text.

In Fig S9, which was performed according to the comment of the reviewer #1, we obtained the similar result even when we used 293 cells expressing CSF1R alone and 293 cells co-expressing CSF1R and TNFAIP2.

Reviewer #3

M-Sec is a 74 kDa protein expressed in myeloid cells of both mouse and human origin, including dendritic cells and macrophages, and in a subtype of enterocytes (M cells) that has homology to a component of the exocyst complex, Sec6. The corresponding gene was identified in 1994 as a tumor necrosis factor-alpha-inducible primary response gene (Tnfaip2) and designated also as B94. This protein was involved in the formation of membrane protrusions extending from the plasma membrane, including tunneling nanotubes. The manuscript by Randa A Abdelnaser et al identifies M-Sec as a partner of Colony Stimulating Factor 1 receptor (Csf1r, also known as Fms) that favors its activation through the formation of aggregates in macrophages, the two proteins interacting with PIP2 (phosphatidylinositol 4,5-biphosphate), which is supposed to drive the formation of aggregates. This limited study, almost exclusively performed in cell lines, is nevertheless clearly reported and the proposed conclusions summarize the results obtained in cell lines. The description of M-Sec /PIP2 pathway to Csf1r signaling, would have been more significant by analyzing the phenotype of M-Sec KO mice. The discussion should better address the potential significance of the described regulation.

Please notice the following changes in the new version:

- (1) We used "CSF1R" in place of "Fms" and "TNFAIP2" in place of "M-Sec" in the new version according to the comment of the reviewer #1.
- (2) The previous Figs 5, 6, 7, 8, 9 and 10 are Figs 4, 5, 6, 7, 8 and 9 in the new version because Figs 3 and 4 were combined according to the comment of the reviewer #2.
- (3) We used "CSF-1-expanded self-renewing bone marrow-derived macrophages" (CSF-1-expanded self-renewing BM-MΦ; Reference #34) in place of "bone-marrow-derived macrophages" to more clearly specify the cells that we employed in this study.

Main comment 1. A key question that cannot be addressed in this study is the functional significance of Csf1r aggregates. As described, Csf1-induced signaling is inhibited by deletion or inhibition of M-Sec. Homozygous M-Sec-KO mice are viable and grow normally, nevertheless develop renal functional alterations worsening with age (PMID: 33722931). Inhibition of Csf1r pathway, for example with therapeutic antibodies, leads to relatively rapid depletion of most tissue macrophage populations and osteoclasts. What is the functional importance of the M-Sec involving pathway? For example, do monocytes that survive and differentiate in liquid culture upon Csf1 exposure die upon inhibition of M-Sec?

Reply

According to the comment, we performed additional experiments using human peripheral blood monocytes and monocyte-derived macrophages, the latter of which were prepared by culturing monocytes for 5 days in the presence of 100 ng/ml CSF-1. As shown in Fig 10, the CSF1R aggregates were hardly detected in monocytes (Fig 10A), but readily observed in monocyte-derived macrophages (Fig 10B), which was presumably due to a higher expression of both CSF1R and TNFAIP2 in monocyte-derived macrophages (Fig S10). Furthermore, the TNFAIP2 inhibitor did not affect the CSF-1-mediated survival of monocytes (Fig 10C), but reduced the CSF-1-mediated survival of monocyte-derived macrophages (Fig 10D, upper). Such reduction was not observed for the CSF-2-mediated survival of monocyte-derived macrophages (Fig 10D, lower). These results suggest that TNFAIP2 functions as the regulator of CSF1R in macrophages, but not in undifferentiated monocytes. We appreciate the comment. The experimental procedures of Figs 10 and S10 were described in the legends, and the Materials and Methods section (page 18, lines 18 - 26, / page 23, lines 7 - 15). To explain and discuss the results in Figs 10 and S10, we added the following paragraph and sentences (underlined):

[Page 12, line 17 - page 13, line 2 (Results)]

Human peripheral blood monocyte-derived macrophages form CSF1R aggregates, and TNFAIP2 inhibition reduces CSF-1-mediated survival of monocyte-derived macrophages

Finally, we performed the experiments using human peripheral blood monocytes and monocyte-derived macrophages, the latter of which were prepared by culturing monocytes for 5 days in the presence of CSF-1 (49). The CSF1R aggregates were hardly detected in monocytes (Fig 10A), but readily observed in monocyte-derived macrophages (Fig 10B), which was presumably due to a higher expression of both CSF1R and TNFAIP2 in monocyte-derived macrophages (Fig S10). Furthermore, the TNFAIP2 inhibitor did not affect the CSF-1-mediated survival of monocytes (Fig 10C), but reduced the CSF-1-mediated survival of monocyte-derived macrophages (Fig 10D, upper). Such reduction was not observed for the CSF-2-mediated survival of monocyte-derived macrophages (Fig 10D, lower). These results suggest that TNFAIP2 functions as the regulator of CSF1R in macrophages, but not in undifferentiated monocytes. This idea is further supported by the experiments using the CSF-1-expanded self-renewing mouse bone marrow-derived macrophages and the mouse macrophage cell line RAW264.7 cells.

[Page 16, line 26 - page 17, line 2 (Discussion)]

The present study suggests that TNFAIP2 functions as the regulator of CSF1R in macrophages, but not in undifferentiated monocytes. Therefore, it will be interesting to analyze how monocytes and tissue-resident macrophages are differently affected by the

TNFAIP2 knockout in mice.

Main comment 2-1. References 21 to 23 are mentioned but these manuscripts describe Csf1r interactions with other cellular components in response to the ligand or in the nucleus but do not mention such aggregates. Can we observe such aggregates in monocytes as well? in cancer cells expressing Csf1r? Are these aggregates an absolute requirement for Csf1 signaling?

Reply

Modified from Figure 1 of Reference #22

The biochemical analysis of Reference #21 raised the possibility that CSF1R monomers form aggregates in the mouse macrophage cell line BAC1.2F5. The formation of CSF1R aggregates was not the main topic of References #22 and #23, but the immunofluorescence of these studies showed that CSF1R was detected as aggregate-like signals in BAC1.2F5 cells, as shown above (the aggregate-like CSF1R signals are indicated by yellow arrows). To more accurately explain these points, we rewrote the sentences as follows (underlined):

[Page 4, lines 6 - 9 (Introduction)]

Interestingly, the biochemical analysis raised the possibility that CSF1R monomers are clustered and form aggregates in the mouse macrophage cell line BAC1.2F5 (21). Consistent with this, CSF1R was detected as aggregate-like signals in BAC1.2F5 cells in the immunofluorescence analysis (22, 23).

Main comment 2-2. References 21 to 23 are mentioned but these manuscripts describe Csf1r interactions with other cellular components in response to the ligand or in the nucleus but do not mention such aggregates. Can we observe such aggregates in monocytes as well? in cancer cells expressing Csf1r? Are these aggregates an absolute requirement for Csf1 signaling?

Reply

(1) As shown in Fig 10, such aggregates were hardly detected in monocytes, but readily observed in monocyte-derived macrophages. We explained and discuss the results in this new version (please see our reply to Main comment-1).

(2) Reference #23 showed that CSF1R expressed in the breast cancer cell line SKBR3 did not form aggregates, but CSF-1 could stimulate their proliferation. Thus, the formation of CSF1R aggregates may not be an absolute requirement for its ligands to activate CSF1R. However, as proposed by Reference #21, the pre-formed CSF1R aggregates, in which the monomers are close to each other in macrophages, may be beneficial for CSF-1 or IL-34 to dimerize and activate CSF1R. The present study supports the idea. To clarify the point, we added the following sentences and words (underlined):

[Page 4, lines 6 - 14 (Introduction)]

Interestingly, the biochemical analysis raised the possibility that CSF1R monomers are clustered and form aggregates in the mouse macrophage cell line BAC1.2F5 (21). Consistent with this, CSF1R was detected as aggregate-like signals in BAC1.2F5 cells in the immunofluorescence analysis (22, 23). Meanwhile, CSF1R expressed in the breast cancer cell line SKBR3 did not form such aggregates, but CSF-1 could stimulate their proliferation (23). Thus, the formation of CSF1R aggregates may not be an absolute requirement for its ligands to activate CSF1R, but the pre-formed CSF1R aggregates, in which the monomers are close to each other in macrophages, may be beneficial for CSF-1 or IL-34 to dimerize and activate CSF1R (21).

Minor comment 1. The introduction should better prepare the reader by describing M-Sec discovery, functions, manipulation (including the phenotype of ko mice) and inhibition in more details.

Reply

According to the suggestion, we added the following sentences (underlined):

[Page 4, line 20 - page 5, line 6 (Introduction)]

TNFAIP2, which was initially identified as a TNF- α -inducible gene in endothelial cells (24), is highly expressed in hematopoietic tissues (25) and enriched in myeloid cells, such as neutrophils, dendritic cells, monocytes and macrophages (26). TNFAIP2 is the 74-kDa cytosolic protein with no known enzymatic activity and shares a homology with Sec6 (26), a component of the exocyst complex. The well-known function of TNFAIP2 is the formation of tunneling nanotubes (26, 27), the F-actin-containing long plasma membrane protrusions. TNFAIP2 is also known to enhance cell motility (28, 29). Interestingly, podocytes, the glomerular visceral epithelial cells, also constitutively express TNFAIP2.

and the TNFAIP2 knockout mice develop focal segmental glomerulosclerosis (30). In addition, we have demonstrated that TNFAIP2 facilitates cell-to-cell transmission of human retroviruses, such as HIV-1 and HTLV-1 (31 - 33). For instance, the TNFAIP2 inhibitor NPD3064, a small chemical that inhibits TNFAIP2-mediated tunneling nanotube formation (31), reduced the production of HIV-1 in macrophages (31).

Minor comment 2-1. The name and concentration of the tested M-Sec inhibitor as well as duration of cell treatment should be indicated in both the results section and the legend of Figure 1. Do the cell die at later time point upon exposure? Did the author test a dose-dependent effect?

Reply

According to the comment, we added the information to the Result section and the legends of Figs 1A/B/C, 2A, 4A, 10 and S6.

Minor comment 2-2. The name and concentration of the tested M-Sec inhibitor as well as duration of cell treatment should be indicated in both the results section and the legend of Figure 1. Do the cell die at later time point upon exposure? Did the author test a dose-dependent effect?

According to the comment, we performed an additional experiment using the CSF-1-expanded self-renewing BM-M Φ . As shown in Fig S6, the TNFAIP2 inhibitor reduced the CSF-1-mediated proliferation from day 4 onwards in a dose-dependent manner (upper), but did not affect the CSF-2-mediated survival (lower). The result is quite similar to that of the experiment using human monocyte-derived macrophages (Fig 10D; please see our reply to Main comment-1). The experimental procedures of Fig S6 were described in the legend. To explain the result in Fig S6, we added the following sentences (underlined):

[Page 7, lines 2 - 7 (Results)]

The CSF-1-expanded self-renewing BM-M Φ proliferated in the presence of CSF-1, but not of CSF-2 (34), presumably because of their adaptation to CSF-1 during the long-term culture with CSF-1. However, they could survive even in the presence of CSF-2 alone (34). When added to the cultures containing CSF-1 or CSF-2, the TNFAIP2 inhibitor reduced the CSF-1-mediated proliferation from day 4 onwards, but did not affect the CSF-2-mediated survival (Fig S6).

Minor comment 3. Figure 2D, M-Sec knock-down appears to significantly sensitize macrophages to CSF-2-driven phagocytosis. The authors do not mention nor comment

this observation that correlates with an increased signaling on Figure 3 (p-ERK and p-p38 at 1 min).

Reply

We are sorry for our insufficient explanation. The TNFAIP2 knockdown RAW264.7 cells showed a higher basal level of phagocytic activity than the control cells (Fig 2D, please compare the control and TNFAIP2 knockdown cells at "0 h"). Such enhanced phagocytic activity was also observed for the TNFAIP2 inhibitor-treated CSF-1-expanded self-renewing BM-M Φ (data not shown), suggesting that TNFAIP2 negatively regulates phagocytic activity of macrophages although the underlying mechanism remains unexplored. As shown (Fig 2D, left), CSF-1 enhanced the phagocytic activity in the control RAW264.7 cells, but only slightly in the TNFAIP2 knockdown cells despite their higher basal activity. In a sharp contrast, CSF-2 enhanced the phagocytic activity in both the control and TNFAIP2 knockdown cells regardless of their different basal phagocytic activities (Fig 2D, right). To clarify the point, we rewrote the explanation sentences as follows (underlined):

[Page 7, lines 23 - page 8, line 2 (Results)]

The TNFAIP2 knockdown cells showed a higher basal level of phagocytic activity than the control cells (Fig 2D, "0 h"), for unknown reasons. When added to these cells, CSF-1 enhanced the phagocytic activity in the control cells, but only slightly in the TNFAIP2 knockdown cells despite their higher basal phagocytic activity (Fig 2D, left). In a sharp contrast, CSF-2 enhanced the phagocytic activity in both the control and TNFAIP2 knockdown cells regardless of their different levels of basal phagocytic activity (Fig 2D, right).

Minor comment 4. It would be useful to indicate to which extent the formation of Fms aggregates is a characteristic of any cell type expressing Csflr/Fms.

Reply

In the previous version, we showed that the CSF-1-expanded self-renewing mouse bone marrow-derived macrophages and the macrophage cell line RAW264.7 cells formed CSF1R aggregates. In this version, we newly showed that human monocyte-derived macrophages, but not monocytes, formed CSF1R aggregates (Fig 10; please see our reply to Main comment 1). We also explained that mouse macrophage cell line BAC1.2F5 (reference #22), but not the breast cancer cell line SKBR3 (Reference #23), formed CSF1R aggregates (the Introduction section; please see our replies to Main comments 2-1 and 2-2).

February 4, 2025

RE: Life Science Alliance Manuscript #LSA-2024-03032-TR-A

Prof. Shinya Suzu
Kumamoto University
Center for AIDS Research
2-2-1 Honjo
Kumamoto, Kumamoto 860-0811
Japan

Dear Dr. Suzu,

Thank you for submitting your revised manuscript entitled "Identification of TNFAIP2 as a unique cellular regulator of CSF-1 receptor activation". We would be happy to publish your paper in Life Science Alliance pending final revisions necessary to meet our formatting guidelines.

- please be sure that the authorship listing and order is correct
- please upload all figure files as individual ones, including the supplementary figure files
- please add the Twitter and Bluesky handles of your host institute/organization as well as your own or/and one of the authors in our system
- please consult our manuscript preparation guidelines <https://www.life-science-alliance.org/manuscript-prep> and make sure your manuscript sections are in the correct order
- please add your main and supplementary figure legends to the main manuscript text after the references section;
- please add callouts for Figures 2B and S9A to your main manuscript text

FIGURE CHECK:

- please add sizes next to all blots

LSA now encourages authors to provide a 30-60 second video where the study is briefly explained. We will use these videos on social media to promote the published paper and the presenting author (for examples, see <https://docs.google.com/document/d/1-UWCfbE4pGcDdcgzcmiuJl2XMBJnxKYeqRvLLrLS08s/edit?usp=sharing>). Corresponding or first-authors are welcome to submit the video. Please submit only one video per manuscript. The video can be emailed to contact@life-science-alliance.org

A. FINAL FILES:

B. MANUSCRIPT ORGANIZATION AND FORMATTING:

Sincerely,

Reviewer #1 (Comments to the Authors (Required)):

The authors have addressed my comments. I have concerns that the phenomenon reported is an artefact of the in vitro systems used and may not be relevant in primary macrophages in vivo.

Reviewer #2 (Comments to the Authors (Required)):

This manuscript shows that TNFAIP2 is important regulator of CSF-1R activation in macrophages, explaining earlier observations and elucidating how TNFAIP2 exerts its effects. In my opinion, the authors have adequately addressed the comments of all reviewers and the data strongly support each of the claims made in the manuscript. The authors' attention is drawn to the following points:

- 1) TNFAIP2 should be defined in the abstract
- 2) P9, line 208, change "associates with" to "is associated with"
- 3) Penultimate paragraph. It should be made clear that the diffusion of most of the TNFAIP2 in the cytoplasm doesn't preclude the importance of the demonstration of its selective association with the CSF-1R that they have clearly shown. It seems very likely that the selective association significantly contributes to the formation of CSF-1R aggregates.

Reviewer #3 (Comments to the Authors (Required)):

The authors considered the requested information and suggestion made in the initial review. No additional suggestion.

Reviewer #2

This manuscript shows that TNFAIP2 is important regulator of CSF-1R activation in macrophages, explaining earlier observations and elucidating how TNFAIP2 exerts its effects. In my opinion, the authors have adequately addressed the comments of all reviewers and the data strongly support each of the claims made in the manuscript. The authors' attention is drawn to the following points:

1) TNFAIP2 should be defined in the abstract

Reply

We added the following words (underlined):

Here, we identify the cellular protein TNF- α -induced protein 2 (TNFAIP2) as a unique regulator of CSF1R.

2) P9, line 208, change "associates with" to "is associated with"

Reply

We changed "associates with" to "is associated with".

3) Penultimate paragraph. It should be made clear that the diffusion of most of the TNFAIP2 in the cytoplasm doesn't preclude the importance of the demonstration of its selective association with the CSF-1R that they have clearly shown. It seems very likely that the selective association significantly contributes to the formation of CSF-1R aggregates.

Reply

According to your thoughtful comment, we modified the paragraph as follows (underlined):

It is possible that the formation of CSF1R aggregates by TNFAIP2 is due to their interaction. In our co-immunoprecipitation using TNFAIP2-expressing CSF1R-transfected 293 cells, the wild-type CSF1R and the 1-545 mutant, but not the 1-537 mutant, were detected in the anti-TNFAIP2 immunoprecipitates (Fig S11), which was consistent with the finding that the wild-type CSF1R and the 1-545 mutant, but not the 1-537 mutant, formed aggregates in the presence of TNFAIP2. However, it should be mentioned that TNFAIP2 does not necessarily co-localize with CSF1R since it diffusely localizes throughout the cytoplasm in the transfected 293 cells (Fig S12). The diffuse localization of TNFAIP2 does not preclude the importance for its interaction with CSF1R, but further studies are necessary to understand to what degree the possible TNFAIP2 - CSF1R interaction contributes to the formation of CSF1R aggregates by TNFAIP2. It is also necessary to clarify whether TNFAIP2 directly or indirectly interacts with CSF1R.

February 5, 2025

RE: Life Science Alliance Manuscript #LSA-2024-03032-TRR

Prof. Shinya Suzu
Kumamoto University
Center for AIDS Research
2-2-1 Honjo
Kumamoto, Kumamoto 860-0811
Japan

Dear Dr. Suzu,

Thank you for submitting your Research Article entitled "Identification of TNFAIP2 as a unique cellular regulator of CSF-1 receptor activation". It is a pleasure to let you know that your manuscript is now accepted for publication in Life Science Alliance. Congratulations on this interesting work.

DISTRIBUTION OF MATERIALS:

Again, congratulations on a very nice paper. I hope you found the review process to be constructive and are pleased with how the manuscript was handled editorially. We look forward to future exciting submissions from your lab.

Sincerely,
